# Automated Model Discovery via Multi-modal & Multi-step Pipeline

Lee Jung-Mok[1]    Nam Hyeon-Woo[1]    Moon Ye-Bin [1]    Junhyun Nam[2]    Tae-Hyun Oh[3*]

[1] Dept. of Electrical Engineering, POSTECH
[2] Samsung Electronics
[3] School of Computing, KAIST
{jungmok,hyeonw.nam,ybmoon}@postech.ac.kr,
junh.nam@samsung.com, taehyun.oh@kaist.ac.kr

## Abstract

Automated model discovery is the process of automatically searching and identifying the most appropriate model for a given dataset over a large combinatorial search space. Existing approaches, however, often face challenges in balancing the capture of fine-grained details with ensuring generalizability beyond training data regimes with a reasonable model complexity. In this paper, we present a multi-modal & multi-step pipeline for effective automated model discovery. Our approach leverages two vision-language-based modules (VLM), AnalyzerVLM and EvaluatorVLM, for effective model proposal and evaluation in an agentic way. AnalyzerVLM autonomously plans and executes multi-step analyses to propose effective candidate models. EvaluatorVLM assesses the candidate models both quantitatively and perceptually, regarding the fitness for local details and the generalibility for overall trends. Our results demonstrate that our pipeline effectively discovers models that capture fine details and ensure strong generalizability. Additionally, extensive ablation studies show that both multi-modality and multi-step reasoning play crucial roles in discovering favorable models.

## 1 Introduction

Model discovery aims to identify the optimal model structure and parameters that best represent given data. Historically, model discovery has been conducted manually by scientists [40, 41] and has established groundbreaking advances in science and technology that have shaped our understanding of the world. However, as the complexity and scale of modern datasets continue to grow, the feasibility of manual model discovery has become increasingly limited [11, 46, 58]. Automating this process [7, 14, 23, 33, 38, 46] offers the potential to accelerate the scientific progress by reducing reliance on human experts and efficient exploring of complex model spaces.

Nevertheless, automatically finding the appropriate model structure is inherently challenging as it requires: 1) exploring over a vast combinatorial search space of candidate models, and 2) balancing between interpretability and model fit, ensuring the model captures the data accurately while remaining sufficiently simple and understandable to domain experts. Most existing systems [8, 14, 29, 33, 43] were carefully designed to address these challenges. For instance, [14] and [33] proposed a predefined grammar for kernel composition within Gaussian processes, which structured the search process and reduced the manual effort required to determine kernel composition. Also, the objective function relied on predefined quantifiable metrics for the model selection, rather than dynamically adapting

---

*Corresponding Author

39th Conference on Neural Information Processing Systems (NeurIPS 2025).

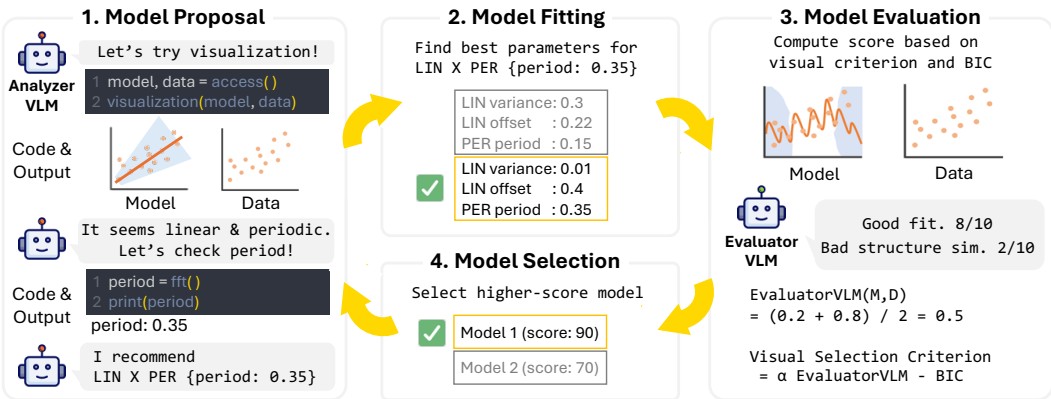

Figure 1: **Overall pipeline.** Our multi-modal & multi-step pipeline contains 4 stages: model proposal, model fitting, model evaluation, and model selection. During the model proposal, AnalyzerVLM repeatedly analyzes the data until it determines the results are sufficient. Then, it proposes a model structure with linearity(LIN) and periodicity(PER), then the model fitting is conducted. In the model evaluation stage, EvaluatorVLM assesses models visually and utilizes the score in the subsequent model selection process.

criteria as a human expert would. However, designing the balanced and sophiscate procedure for composing models requires significant modeling expertise, which compromises the flexibility needed for automation.

To achieve a more flexible and intelligent model discovery system, we substitute human experts' roles in the model discovery with multi-modal agents. First, we introduce AnalyzerVLM, an agent that is capable of conducting in-depth analysis of given data and models through multi-step reasoning. We task AnalyzerVLM with generating code for analyzing the current model and proposing improved candidate models based on its findings. We observe that AnalyzerVLM actively uses existing libraries such as NumPy [42] and Matplotlib [21], which are commonly used by humans for analysis and visual inspection of plots. AnalyzerVLM iteratively analyzes the current model and data until it identifies candidate models that can outperform the given model.

Second, we propose a Visual Information Criterion (VIC) performed by EvaluatorVLM. Our new criterion is designed to incorporate the way humans perceive a given model. Traditional criteria, *e.g.*, Bayesian Information Criterion (BIC), quantify both goodness-of-fits and model complexity to identify undesirable models, but we found that they are often counter-intuitive and fail to consider inherent data characteristics and relationships. On the other hand, humans easily identify inherent data and model trends, structures, and patterns through visualization. By introducing visual representation into the model selection criteria, our VIC complements traditional criteria, enabling the discovered model to escape suboptimal. By alternating the AnalyzerVLM and EvaluatorVLM, we demonstrate that our multi-step and multi-modal pipeline effectively searches over the model space like a human, enhancing the proposed model's quality and showing high generalizability compared to other model discovery methods.

## 2 Related Work

**Automated Model Discovery.** Model discovery typically includes tasks such as selecting the model structure, estimating parameters, validating the model's effectiveness, and refining the model. The discovery process requires extensive expertise across multiple data domains and involves multiple steps. Thus, model discoveries require human experts to repeatedly refine the model and reflect the data domain's prior knowledge. To alleviate these difficulties, automated model discovery processes have been proposed. The goal is to reduce the burden of needing human experts and prior work has been developed by defining model space on Gaussian process [4, 6, 14, 15, 22, 26, 27, 33, 35]. Such methods are based on kernel structure search with pre-defined grammar, iteratively evolving its kernel structure. Automatic model discovery pipeline often uses the greedy-search based approaches [14, 33], sampling-based approaches [44, 45], and employing large language models (LLMs) [30]. Also there are similar approaches in symbolic regression [2, 5, 19, 25, 28, 31, 59, 60, 69], which searches

for the function composition that most fits to the data. Given a basis function and the operators, it automatically finds out the appropriate function composition over evolving. Symbolic regression requires diverse function candidates to be generated and takes the evolving over the generation, it mostly utilizes genetic programming [2, 19, 60, 69], or neural networks such as transformers [5, 24, 31, 55]. These days symbolic regression utilizing LLMs [16, 39, 48, 51]. Thus, utilizing LLMs is promising in the automatic model discovery pipeline, as LLMs can be adapted to those with broad expert knowledge by providing a suitable prompt. In this work, we extend the usage of LLMs to VLMs by effectively harnessing the reasoning, planning, and evaluating capabilities with visual modality.

**LLMs & VLMs-based Data Understanding.** Interpreting and understanding current data is crucial for model discovery and prediction, as extrapolation regimes including future events often depend on past patterns. [17] show that LLM has the ability to effectively deal with time-series data if the numeric tokens are properly designed. [13] propose to use plot visualizations instead of lengthy texts. [30] simply investigate the potential of visual plots as a replacement of text representations to reduce the number of tokens, and shows the comparable alternative to the text inputs. Like such methods for understanding data, there are also automatic data-driven discovery frameworks [37, 70] which automatically analyzes and finds the relation between variables of the data. Our method leverages VLMs to gain a deeper understanding of the given data within the context of model discovery. We let VLM identify and understand visual data itself by effectively overviewing the trends and relationships.

**Model Agents.** LLMs and VLMs, such as GPT-4o [1], have high reasoning and generalization capabilities. These strengths make them particularly well-suited for use as agents in complex tasks [10, 20, 49, 57, 61, 62, 65–68]. Many recent works have utilized LLMs and VLMs for data analysis [20, 53, 63, 70], as well as LLM/VLM for evaluator [9, 18, 32]. In addition, multi-step reasoning techniques [61, 62, 65] are commonly employed to enhance the decision-making and analysis capabilities of LLM-based agents. We leverage this advanced capability of VLMs to accelerate and enhance the model discovery process, facilitating the efficient identification of more robust and accurate models. Specifically, VLMs analyze the data and candidate models by adaptively generating codes to identify potential improvements and evaluate model accuracy through the interpretation of visual plots. By integrating these components, our approach fully capitalizes on the strengths of VLMs to optimize the model discovery workflow.

## 3 Method

### 3.1 Overview

Automated model discovery aims to identify a model structure $\mathcal{M}$ and the corresponding parameters $\theta$ that best describe a dataset $\mathcal{D}$. This process systematically identifies optimal model structures by exploring a structured search space while balancing complexity and fitness. Our pipeline discovers a proper model iteratively, and it has 4 steps in each round: model (1) proposal, (2) fitting, (3) evaluation, and (4) selection, as shown in Fig. 1 and Algorithm 1.

The model proposal step of the $r$-th round needs to suggest better model candidates $\mathcal{M}^r = \{\mathcal{M}_1^r, ..., \mathcal{M}_n^r\}$, given data $\mathcal{D}$ and previous models $\mathcal{M}^{r-1}$. To effectively suggest candidates, we propose an AnalyzerVLM which is designed to propose model candidates through multi-step analysis. Further details about AnalyzerVLM can be found in Sec. 3.2.

Once the candidate models are suggested, they are conveyed to the model fitting step, where we determine the optimal model parameters through marginal likelihood-based optimization. To ensure robustness, we conduct the parameter optimization with multiple initialization points with AnalyzerVLM proposal, incorporating random restarts and inheritance from model candi-

---

**Algorithm 1** Model Discovery Pipeline.

---
1: **Input:** dataset $\mathcal{D}$, rounds $R$, model pool $\mathcal{P}$
2: **Initialize:** best model $\mathcal{M}^*$,
3: **for** $r = 1$ **to** $R$ **do**
4:     $\mathcal{M}^r = \text{AnalyzerVLM}(\mathcal{M}^*, \mathcal{D})$ ▷ *Proposal*

5:     **for** $\mathcal{M} \in \mathcal{M}^r$ **do**
6:         $\theta^* = \text{Optimize}(\mathcal{M}, \mathcal{D})$     ▷ *Fitting*
7:         $s_\mathcal{M} = \alpha \cdot \text{EvaluatorVLM}(\mathcal{M}, \theta^*, \mathcal{D}) - \text{BIC}$     ▷ *Evaluation*
8:     **end for**
9:     $\mathcal{P} \leftarrow \mathcal{P} \cup \mathcal{M}^r$
10:     $\mathcal{M}^* \leftarrow \arg\max_{\mathcal{M} \in \mathcal{P}} s_\mathcal{M}$     ▷ *Selection*
11: **end for**

---

dates [14, 27, 33]. The parameter with the highest likelihood is then selected as the best for subsequent model evaluation.

The fitted model is evaluated using Visual Information Criterion (VIC). To assess the model in multiple aspects and perceptually plausible ways, we propose EvaluatorVLM that measures visual scores: visual fitness and generalizability. The visual score is combined with the traditional measure, Bayesian Information Criterion (BIC). These criteria allow us to assess the model both in detail and holistically. Additional information about our visual scoring is provided in Sec. 3.3.

Finally, scored models are put into the model pool $\mathcal{P}$. The model pool is then sorted based on the VIC. In the next round, reference models are sampled from the updated pool. This round is iterated until the number of iterations reaches the maximum criterion.

## 3.2 AnalyzerVLM: Multi-step Analysis

The reasoning process of AnalyzerVLM can be formalized as a sequential decision making task defined by a policy $\pi(a_t|c_t)$, where $c_t = (o_1, a_1, \cdots, o_{t-1}, a_{t-1}, o_t)$ represents the context, $a_t \in \mathcal{A}$ denotes the action, and $o_t \in \mathcal{O}$ is the observation. Our multi-step analysis integrates the interaction process through the code-execution, getting an observation for the executed output. When there is no observation (when only reasoning happens), the context updates simply as $c_{i+1} = (c_i, a_i)$. When code executions are available, the action space expands to $\mathcal{A} = \mathcal{L} \cup \mathcal{C}$, enabling the agent to execute tool-related actions through the code execution. If the agent selects an action $a_i \in \mathcal{C}$, the interaction with the environment happens, producing an observation $o_i = \text{Execute}(a_i)$. This observation is appended to the context, resulting in the updated context $c_{i+1} = (c_i, a_i, o_i)$.

---

**Algorithm 2** Model Proposal with AnalyzerVLM

1: **Input:** prompt $P$, model $\mathcal{M}$, dataset $\mathcal{D}$, AnalyzerVLM, max context length $N_{\max}$
2: **Initialize:** $c = (P, \mathcal{D}, \mathcal{M})$, step count $i$
3: **while** $|c_i| < N_{\max}$ **do**
4:     $a_i = \text{AnalyzerVLM}(c_i)$
5:     **if** $a_i \in \mathcal{L}$ **then**
6:        $c_{i+1} \leftarrow (c_i, a_i)$          ▷ *Analyze*
7:     **else if** $a_i \in \mathcal{C}$ **then**
8:        $o_i = \text{Execute}(a_i)$       ▷ *Execute*
9:        $c_{i+1} \leftarrow (c_i, a_i, o_i)$
10:    **else if** $a \in 2^{\Sigma}$ **then**
11:       **return** $a_i$            ▷ *Propose*
12:    **end if**
13:    $i = i + 1$
14: **end while**

---

At the initial step, AnalyzerVLM is provided with a prompt $P$ that specifies a task and an objective, a dataset $\mathcal{D}$, and candidate models $\mathcal{P}$ from the previous rounds. These inputs define the initial context $c_1$. Starting from $c_1$, AnalyzerVLM iteratively selects actions $a_i$ based on a fixed policy $\pi(a_i|c_i)$ and finally produces $a_T \subset \Sigma$, which represents the candidate models for the next stage of model discovery. The overall algorithm is shown in Algorithm 2. The action space of AnalyzerVLM consists of three subspaces: a language space $\mathcal{L}$ for the natural language reasoning process, a code space $\mathcal{C}$ for generating executable code to perform analysis, and a model space $2^{\Sigma}$ for generating candidate models. Since AnalyzerVLM proposes multiple candidate models, the model space is represented as a power set of the search space $\Sigma$. Each specific action is described as follows.

**Analyze.** When AnalyzerVLM performs analysis at $i$-th step (*i.e.*, $a_i \in \mathcal{L}$), it means that it will generate the analysis and formulate a new plan for the next step in natural language, based on the current context $c_i$. As the analysis and planning are conducted entirely in natural language, the resulting analysis can be directly incorporated into the next context $c_{i+1} = (c_i, a_i)$.

**Execute.** When AnalyzerVLM chooses to execute code (*i.e.*, $a_i \in \mathcal{C}$), it generates executable code block in the python language. The generated code block is executed, and the resulting observation $o_i = \text{Execute}(a_i)$ is used to update the context for the next step $c_{i+1} = (c_i, a_i, o_i)$. We highlight that the observation $o_i$ is not limited to textual or numeric outputs, but can also include visual representations (*e.g.*, plots), enabling AnalyzerVLM to perform the diverse analysis of the given data into the most-fittable format for it.

**Propose.** When AnalyzerVLM has sufficiently analyzed the data and model, AnalyzerVLM chooses to perform model proposal based on the previous contexts $c_i$ (*e.g.*, analysis) . When the propose action is conducted, the multi-step analysis is terminated. When AnalyzerVLM proposes the model structure, it also propose the initial parameters based on its analysis. Based on its iterative analysis, it

can effectively propose the initial parameters for the model structure (*e.g.*, period), and utilizing those parameters properly can enhance the good structure's good fitting.

Compared to the BoxLM [30] proposes the candidate model at once, our AnalyzerVLM utilizes multi-step reasoning that acquires the sufficient information by itself to propose the model. Also, our AnalyzerVLM dynamically chooses the way to look at the data and model, giving a degree of freedom to AnalyzerVLM for deep analsyis. With the multi-step pipeline, our model suggestion can effectively identifies missing characteristics in the current candidate model in various ways.

### 3.3 EvaluatorVLM: Visual Information Criterion

From a Bayesian perspective, model selection is grounded in the marginal likelihood, which measures how well a model explains the observed data while integrating over all possible parameter values:

$$p(\mathcal{D}|\mathcal{M}) = \int p(\mathcal{D}|\mathcal{M}, \theta)p(\theta|\mathcal{M})d\theta. \tag{1}$$

The parameter optimization performed in the model fitting step of automated model discovery can be interpreted as a practical surrogate for intractable marginal likelihood computation. A widely adopted approach to approximate marginal likelihood is using Laplace's method around the maximum likelihood estimator $\theta^*$, leading to the Bayesian Information Criterion (BIC):

$$\text{BIC}(\mathcal{M}, \mathcal{D}) = -2\log p(\mathcal{D}|\mathcal{M}, \theta^*) + |\mathcal{M}|\log|\mathcal{D}|, \tag{2}$$

where $|\mathcal{M}|$ is the number of model parameters, and $|\mathcal{D}|$ is the size of the dataset.

While marginal likelihood naturally balances model fit to the train data and complexity, it may struggle to generalize to unseen regions [34]. The limitations of marginal likelihood in evaluating generalization stem from its inability to capture the structural properties of data that persist across both the training and test regions. Inspired by this, we propose a novel model selection criterion by utilizing VLMs in identifying suitable models. Specifically, we feed a visualization of the posterior predictive results to EvaluatorVLM and evaluates how well the model suits the data. This evaluation score is then incorporated into our model selection process with the model evidence.

To integrate VLM-based judgment into the model selection, we propose the *Visual Information Criterion (VIC)*. We define VIC as a weighted combination of the EvaluatorVLM score and BIC:

$$VIC(\mathcal{M}, \mathcal{D}) = \alpha \cdot \text{EvaluatorVLM}(\mathcal{M}, \theta^*, \mathcal{D}) - \text{BIC}. \tag{3}$$

Since BIC is traditionally lower for better models, we negate it to ensure that higher VIC values indicate better models. Despite its simplicity, VIC effectively approximates the posterior model probability $p(\mathcal{M}|\mathcal{D})$ by incorporating an additional prior term $p(\mathcal{M})$, extending beyond the standard BIC-based approach. The detailed derivation of this approximation can be found in Appendix A.2.

Our VIC evaluates two aspects of the model: 1) **Visual Fitness** and 2) **Visual Generalizability**. The reasons of considering both aspects are that 1) visual fitness represents how well the discovered model fits to the data's trend, and 2) visual generalibzaility represents the consistency of trends across training and non-training regions, which can be one way of measuring model's generalization. Each score is measured multiple times and is averaged due to the stochasticity of VLMs. Then VIC is computed by summing up all the following visual scores.

**Visual Fitness.** Visual fitness quantifies how much the data resembles the prediction visually. Specifically, fitness is evaluated through comparing two plots: Given the data observation plot and the predicted posterior mean plot, EvaluatorVLM is prompted to compare two plots to measure the similarity of the prediction and data. We also evaluate visual uncertainty, how large the uncertainty region is given a predicted posterior mean and confidence region. If the uncertainty region is big, we let EvaluatorVLM give a low score, and if the uncertainty region gets suddenly large in non-training data regions, a low score is given. We prompt EvaluatorVLM to score each score in the range of $[0, 50]$. Through the summation of mean prediction resemblance score and the plot uncertainty score, we calculate the visual fitness. The prompts are shown at Appendix A.10.

**Visual Generalizability.** Visual generalizability quantifies whether the predicted posterior mean preserves it structural consistency in extrapolated regions beyond the training data. Given a plot

Table 1: **Quantitative results.** We compare our pipeline with five competing methods on the train and test region, reporting RMSE. On average, our pipeline outperforms the others. **Bold** stands for the best, and underline for the second best.

| Method | Airline Train | Airline Test | Solar Train | Solar Test | Mauna Train | Mauna Test | Wheat Train | Wheat Test | Call Train | Call Test | Radio Train | Radio Test | Gas Train | Gas Test | Avg. Train | Avg. Test |
|---|---|---|---|---|---|---|---|---|---|---|---|---|---|---|---|---|
| Gaussian Process (SE) [47] | 0.0696 | 0.1432 | 0.0334 | 0.6650 | 0.0320 | 0.0782 | 0.0329 | 0.5426 | 0.0386 | 0.3120 | 0.0426 | 0.1835 | 0.0468 | 0.2605 | 0.0423 | 0.3121 |
| ARIMA [52] | 0.2513 | 0.1583 | 0.2400 | 0.3259 | 0.2995 | 0.1788 | 0.1638 | 0.1537 | 0.2786 | 0.6291 | 0.3766 | 0.3828 | 0.3158 | 0.1507 | 0.2751 | 0.2827 |
| Facebook Prophet [54] | 0.2526 | 0.1648 | 0.2208 | 0.3924 | 0.2998 | 0.0846 | 0.1606 | **0.1260** | 0.2676 | 0.8988 | 0.3242 | 0.6408 | 0.3154 | 0.1682 | 0.2630 | 0.3536 |
| Automatic Statistician [14] | **0.0056** | 0.2004 | **0.0265** | 0.6370 | 0.0028 | 0.0725 | 0.0640 | 0.1499 | 0.0279 | 0.8197 | **0.0253** | 0.1505 | 0.0114 | **0.0822** | 0.0234 | 0.3017 |
| BoxLM [30] | 0.0106 | 0.2845 | 0.0312 | 0.4523 | 0.0028 | 0.5312 | 0.0114 | 0.1469 | **0.0073** | 0.2366 | 0.0312 | 0.2975 | **0.0096** | 0.4831 | 0.0149 | 0.3474 |
| Ours (Qwen2.5-VL) | 0.1350 | 0.0469 | 0.1834 | 0.3037 | 0.0826 | 0.0898 | 0.1127 | 0.1595 | 0.2593 | **0.0508** | 0.3443 | 0.0558 | 0.0969 | 0.0786 | 0.1735 | 0.1122 |
| Ours (GPT-4o) | 0.0057 | **0.0369** | 0.0297 | **0.2861** | **0.0025** | **0.0497** | **0.0006** | 0.1316 | 0.0386 | 0.4742 | 0.1346 | 0.0784 | 0.1034 | 0.0843 | 0.0451 | 0.1556 |
| **Ours (GPT-4o-mini)** | 0.0066 | 0.0534 | 0.0297 | **0.2861** | 0.0026 | 0.0564 | 0.0183 | 0.1470 | 0.0093 | **0.0508** | 0.0254 | 0.0562 | **0.0078** | 0.0893 | **0.0134** | **0.1070** |

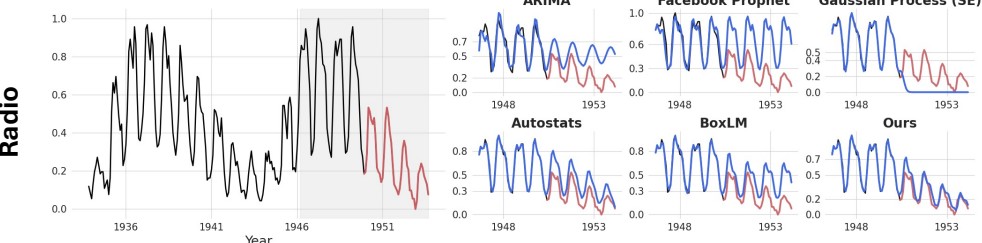

Figure 2: **Qualitative results.** The graph on the left illustrates the data, where a black line stands for the observed data and a red line for the test data. The graphs on the right show the predictions of each method at the shaded region of the left graph, with the blue lines representing their outputs. The results highlight the generalization capabilities and efficient search performance of our method.

of the posterior mean and its confidence region, EvaluatorVLM is prompted[2] to measure whether the model's predictions are maintained in the extrapolated region in the range of $[0, 50]$, without requiring ground-truth labels in extrapolated regions. This allows us to quantify how well the model generalizes beyond the observed data distribution. Specifically, we have visualized the posterior mean and confidence at the 20% extrapolated region for each side, and instructed EvaluatorVLM to check whether the 1) posterior mean flattens 2) confidence region suddenly increases at extrapolated region.

## 4 Experiments

### 4.1 Gaussian Process Kernel Discovery

**Datasets and Competing Methods.** We evaluate our multi-modal & multi-step pipeline on real-world univariate datasets [33], including Airline Passenger, Solar Irradiance, Mauna Loa, Wheat, Call-Center, Radio, and Gas Production. We will refer to the data by their representative terms for convenience, *e.g.*, Solar for Solar Irradiance. And we compare our pipeline against five competing methods ranging from traditional forecasting methods to the latest LLM-based model discovery approaches: Gaussian Process Regression with Squared Exponential kernel, ARIMA [52], Facebook Prophet [54], Automatic Statistician [14, 33], and BoxLM [30][3]. We employ GPT-4o-mini[4] for methods using LLMs, including ours. Additional experimental details (*e.g.*, basis kernels and kernel grammars) are given at the Appendix A.3.

**Result.** As shown in Table Ref tab:quantitative, our method can discover better models by achieving consistently lower RMSE compared to other methods on average. While BoxLM and Gaussian Process (SE) exhibit low RMSE values on the training set, their RMSEs significantly increases in the test region, indicating poor generalization. In contrast, our method maintains consistently low RMSEs across both training and test sets, highlighting its ability to generalize effectively beyond the training data. This shows the robustness of our method in the test region through AnalyzerVLM and EvaluatorVLM.

---

[2]The prompt is shown at the Appendix A.10.

[3]The authors used GPT-4v, but for fair comparison with ours, we update the LLM version with GPT-4o-mini.

[4]Note that our main result refers to the one using GPT-4o-mini. We have varied the VLM into Qwen2.5-VL, GPT-4o, and GPT-4o-mini for the ablation study, shown at Section 4.2.

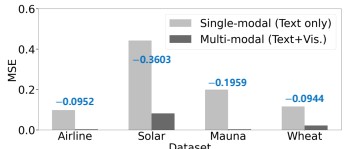 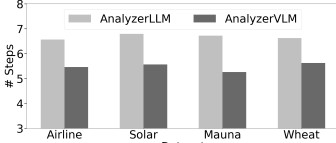 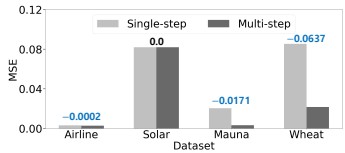

Figure 3: **MSE of text only and multimodal representation.** We restrict the usage of visual representation during model generation and evaluation processes. The blue value indicates the reduction in MSE when using visual representation.

Figure 4: **The required step number of LLM and VLM.** When using LLM instead of VLM for Analyzer, *i.e.*, without visualizing the data and model, Analyzer requires more steps to validate its analysis.

Figure 5: **MSE of the multi-step and single-step restriction.** We limit AnalyzerVLM to a single step. Blue values indicate the reduction in MSE when using multi-step analysis compared to single-step analysis.

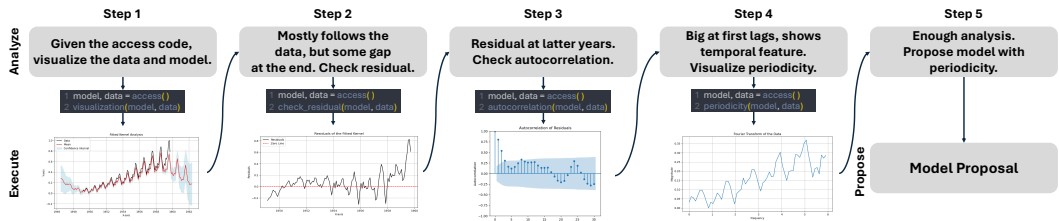

Figure 6: **Multi-step analysis of AnalyzerVLM.** We visualize an example of the AnalyzerVLM's multi-step analysis. AnalyzerVLM begins with the data and model visualization, followed by visualizing residuals and identifying the remaining periodic pattern in the data before the model proposal. Finally, AnalyzerVLM proposes a refined model based on the analyses.

We visualize the model discovery result in Fig. 2. ARIMA struggles to capture trends and periodicity in the test region. The Gaussian Process successfully captures the overall trend in the train region but does not fully account for periodicity and finer details. Similarly, Autostats and BoxLM exhibit slight deviations from the trend and miss some finer patterns. Facebook Prophet effectively captures both trend and periodicity but falls slightly behind our method in capturing finer details. However, our method identifies a plausible model that generalizes well to both observed and unseen test data.[5] To further investigate the reasons behind the strong generalization performance, we conduct an ablation study in the following section.

## 4.2 Ablation Study and Analysis

**AnalyzerVLM and EvaluatorVLM.** We conduct an ablation study on the VLM-based modules in our pipeline to examine the impact of AnalyzerVLM and EvaluatorVLM, as shown in Table 2.

In experiments without AnalyzerVLM, we replace it with a simpler model proposal approach for VLM without analysis, providing just visualizations of the model and data while prompting the model generation following BoxLM [30]. For experiments without EvaluatorVLM, we exclude the visual criterion and only rely on the BIC for model selection. The results in Table 2 show that each component plays a crucial role in discovering better models. Our proposed pipeline with both AnalyzerVLM and EvaluatorVLM finds the best model in most cases.

Table 2: **Ablation study on VLM-based modules in our pipeline.** We conduct an ablation study of AnalyzerVLM and EvaluatorVLM in terms of MSE. The last row stands for our complete pipeline. **Bold** represents the best, and underline is the second best.

| Analyzer VLM | Evaluator VLM | Dataset | | | |
|:---:|:---:|:---:|:---:|:---:|:---:|
| | | Airline | Solar | Mauna | Wheat |
| - | - | 0.0810 | 0.1791 | 0.2820 | 0.0216 |
| - | ✓ | 0.0698 | 0.0820 | 0.2863 | 0.0498 |
| ✓ | - | 0.0029 | 0.0822 | 0.0198 | **0.0135** |
| ✓ | ✓ | **0.0028** | **0.0819** | **0.0032** | 0.0216 |

Also our method can be applied across different models including commercial models(GPT), and open-source models. We have varied the backbone models of AnalyzerVLM and EvaluatorVLM. Table 1 shows that our methods can be applied to diverse VLMs, including Qwen2.5-VL [3], GPT-4o-

---

[5]Additional visualization can be found in Appendix.

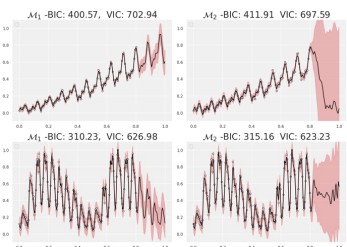

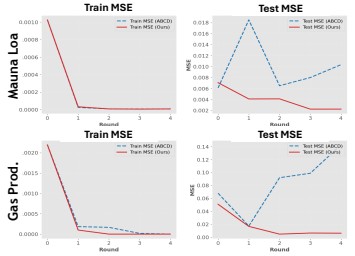

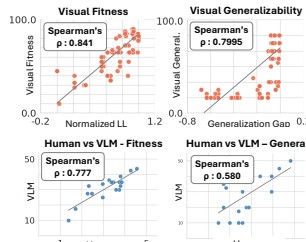

Figure 7: **BIC and VIC comparison.** Comparing BIC and VIC, VIC can sort out the models with low generalizability, by giving them the worse score, while BIC cannot penalize such low generalizability cases.

Figure 8: **MSE of model over rounds.** Train MSE gets lower as round goes, but without VIC, test MSE may increase over rounds. With VIC, it can effectively lower both train and test MSE.

Figure 9: **Correlation of VIC with likelihood, generalization, and human.** At first row, we visualize the correlation between VIC and metrics (likelihood, generalization gap). Second row is human correlation result with VIC.

mini and GPT-4o [1]. Our pipeline can work both with open-source model and commercial models, and it shows that the VLM with high capability has high performance on average.

**Multi-modal vs. Single-modal.** Our pipeline leverages both visual and text representations during model proposal and evaluation. To investigate the impact of the visual representation, we restrict AnalyzerVLM to only use text during the model proposal and rely solely on BIC for model evaluation, not giving the model prediction visualization. The result in Fig. 3 shows that incorporating visual representation improves model discovery performance. Using multimodal representation allows us to consider the overall trend rather than focusing solely on the local values in the text, leading to a better understanding of the data.

**Hyperparameter $\alpha$ selection.** We have set $\alpha$ so that EvaluatorVLM accounts for approximately 10-30% of the original metric (e.g., BIC), and performed grid search around this range to find the best. Since the scales of the selection criteria vary due to data normalization, we set $\alpha$ such that the visual score contributes a consistent proportion to the total objective. We provide the result of test RMSE

| Dataset | $\alpha = 0$ | $\alpha = 30$ | $\alpha = 50$ | $\alpha = 70$ | $\alpha = 100$ |
|---------|--------------|---------------|---------------|---------------|----------------|
| Airline | 0.0937 | 0.0574 | **0.0534** | 0.0612 | 0.0824 |
| Radio | 0.0766 | 0.0715 | **0.0562** | 0.0764 | 0.0954 |

Table 3: **Performance across different $\alpha$ values.** There is a break down point of performance of $\alpha$ around 50, indicating the trade-off between model accuracy and generalizability.

of grid search varying $\alpha$ around 0 to 100, at Airline and Radio dataset at Table 3 As shown, there is a certain turning point (or a break-down point) of alpha around 50. While $\alpha$ is 0, final searched models are dropping at extrapolated region, and when increases, the pipeline tends to select the model with more generality even in high value ranges near 70. This hints that our hyperparameter setting is not sensitive and starting from introducing small $\alpha$ would be a good rule of thumb.

**Reasoning Process of AnalyzerVLM.** AnalyzerVLM proposes a model candidate through the multi-modal and multi-step analysis. To better understand its reasoning process, we compare the number of analysis steps required when using an LLM versus a VLM as the analyzer, as shown in Fig. 4. Both analyzers independently determine when to stop the analysis. AnalyzerLLM requires more steps to generate a proper candidate than AnalyzerVLM. LLM is less effective than VLM at capturing the overall trend, which leads to more iteration for checking its analysis.

Next, we restrict the number of reasoning steps of AnalyzerVLM to compare single-step and multi-step analysis, observing performance improvements across steps, as shown in Fig. 5. The results indicate that additional analysis steps lead to the discovery of a better model. In summary, the overall results demonstrate that the multi-modal and multi-step process enables AnalyzerVLM to make better decisions. The qualitative result of the reasoning process is shown in Fig. 6.

**Generalizability-Aware Model Evaluation.** We show two qualitative examples with VIC($\uparrow$) and -BIC($\uparrow$)[6] scores in Fig. 7. When BIC is superior, it shows deviation at the extrapolated region, but it does not detect such cases and select the right models with the deviation. However, VIC distinguishes such cases, penalizes such models and selects the left models with high generalizability. Specifically at Airline data, as shown in the right-top example of Fig. 7, the model shows a certain drop at the

---

[6]Since BIC is better for the lower value, we negate it for the explanation.

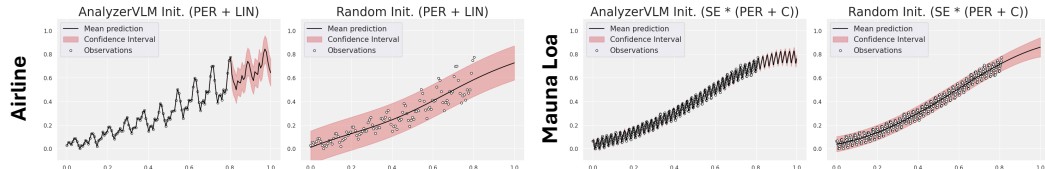

Figure 10: **Visualization of the optimized model with different initializations.** The model optimized from the AnalyzerVLM-proposed initialization reliably identifies appropriate parameter settings through data-driven analysis, whereas random initialization frequently leads to suboptimal configurations.

extrapolated region, which is not a natural behavior. The left-top model shows the natural upward trend at the extrapolated region. BIC does not have any constraints about evaluating such naturalness, it selects out the right model. A similar phenomenon also happens in the Radio data, as shown in bottom example of Fig. 7, the right model's prediction suddenly flattens at the right extrapolate region and the structure is not maintained, which may mean low the generalizability. Penalizing such generalizability, VIC can effectively distinguish such cases. Fig. 8 shows the visualization of the mean squared error of train and test region, for each round. With BIC evaluation, it shows a low train MSE with increasing test MSE. In contrast, our pipeline keeps the test MSE low and even reduces it across rounds, showing better generalization without overfitting.

**Correlation of Visual Information Criterion.** The first row's left image in Fig. 9 shows that our visual fitness is highly correlated to the likelihood, which implies that we can *visually* measure the model's fitness. The first row's right image in Fig. 9 shows the correlation of our visual generalizability and generalization gap. Specifically, we adapt the difference of test MSE and train MSE as the generalization gap. As shown, it shows a high correlation with the generalization gap, which means that our visual generalizability is an effective criteria for evaluating a model's generalization capacity.

The second row of Fig. 9 shows the correlation of VIC's each criterion with the human evaluation. Human evaluation is done by instructing human with the same criteria that EvaluatorVLM: visual fitness and visual generalizability. For human evaluation, we instructed human to rate the given model from 1 to 5. The instruction is shown at Appendix. As shown, the EvaluatorVLM's model evaluation follows a similar trend to the human model evaluation, showing a high correlation at both visual fitness and visual generalizability evaluation. This shows that EvaluatorVLM closely matches human judgment at model evaluation, enabling more efficient and reliable automated assessments of the model discovery.

**Hyperarameter Initialization of AnalyzerVLM.** As reported in [14] and [27], hyperparameter optimization for Gaussian process regression is a non-convex problem, making good initialization crucial for effective model discovery. To address this problem, [14] utilized random initialization with hyperparameter inheritance over rounds, and [27] has utilized random restarts with strong prior, sampling the hyperparameters from certain prior distribution. In our case, we utilize AnalyzerVLM to propose model structures and suggest initialization point based on its analysis. In particular, we initialize the period and lengthscale of the periodic kernel and the lengthscale of the squared exponential kernel, using values suggested by AnalyzerVLM, then start optimizing in the first round. Then, such well-estimated hyperparameter values proposed by AnalyzerVLM can be carried over rounds, enabling the construction of progressively more complex and refined model structures initialized with strong hyperparameter estimates. Figure 10 shows two examples of optimized models with AnalyzerVLM proposal initialization and random initialization. As shown, with the AnalyzerVLM initialization, the model can be optimized to appropriate hyperparameters, while random initialization fails at finding the appropriate hyperparameters and leads to the whole kernel structure's failure.

### 4.3 Application to Symbolic Regression

In this section, we extend our discovery pipeline to symbolic regression, demonstrating its applicability beyond probabilistic model classes. Accordingly, the BIC component of our original VIC is replaced with a commonly used objective in symbolic regression: normalized mean squared error combined with a complexity penalty [50].

**Dataset and Competing Methods.** Extending our model discovery framework into the function discovery, we prompted AnalyzerVLM to generate a normal function rather than the kernels, and

Table 4: **Quantitative results at symbolic regression**. We conduct an experiment on symbolic regression at four datasets: R, Constant, Keijzer, Nguyen. We report R-square and RMSE scores for each dataset. **Bold** represents the best, and underline is the second best. As shown, our method shows competitive results compared to the other methods.

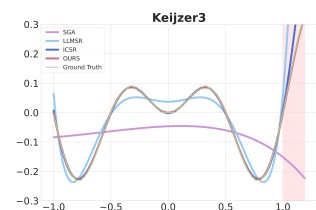

Figure 11: **Qualitative result at symbolic regression.** Our method successfully discovers function composition which fits the data.

| Method | Dataset | | | | | | | |
| --- | --- | --- | --- | --- | --- | --- | --- | --- |
| | R | | Constant | | Keijzer | | Nguyen | |
| | $R^2$ (↑) | RMSE (↓) | $R^2$ (↑) | RMSE (↓) | $R^2$ (↑) | RMSE (↓) | $R^2$ (↑) | RMSE (↓) |
| SGA [36] | 0.8951 | 0.1639 | 0.5677 | 0.1056 | 0.3602 | 0.3263 | 0.8918 | 0.1761 |
| ICSR-V [39] | 0.9808 | 0.1320 | **0.9967** | **0.0209** | 0.9463 | 0.0398 | **0.9952** | 0.0803 |
| LLM-SR [51] | 0.9717 | **0.0805** | 0.9807 | 0.0225 | **0.9972** | **0.0139** | 0.9440 | **0.0240** |
| Ours | **0.9872** | 0.1154 | 0.9503 | 0.0411 | 0.9521 | 0.0362 | 0.9743 | 0.0871 |

utilized our VIC for symbolic regression score. Following [39], we have conducted our experiments at Nguyen [60], Constant [31], R [28], Keijzer [25]. And we compared our methods with ICSR [39], LLM-SR [51], and SGA [36]. Those methods utilize LLM, while ICSR-V also incorporates the visualization of the plot at the function generation phase. The implementation details are shown in Appendix A.3.

**Result.** Our proposed method achieves performance that is competitive with existing approaches on symbolic regression, as shown at Table 4. Our method consistently achieves consistently high predictive accuracy and reliable generalization. Leveraging the AnalyzerVLM and EvaluatorVLM modules, our model discovery pipeline can be employed as a symbolic regressor, enabling interpretable function modeling with our method's iterative analysis and visual evaluation. Fig. 11 shows that our approach successfully captures the underlying functional form on the Keijzer3, closely matching the ground truth in the training region and also showing similar results in extrapolated regions beyond the training data. This shows our generalization performance is robust in capturing the data's structure rather than merely fitting the data.

## 5 Conclusion

We propose a multi-modal multi-step pipeline for automatic model discovery by introducing two VLM-based modules: AnalyzerVLM and EvaluatorVLM. AnalyzerVLM iteratively plans and executes the analysis to propose the most suitable model by analyzing the given data and models. Leveraging large-scale VLM as a multi-modal agent, it can generate analysis and interpret plots and data characteristics in context. Through its multi-step analysis, our method can discover a better model which fits to the data's underlying structure. EvaluatorVLM assesses the suggested model based on the visual representation, *i.e*., plot, evaluating fitness for local details and structure similarity for overall trends. It provides a robust mechanism for model validation beyond numeric error metrics. The experimental results demonstrate that our pipeline effectively discovers a proper model, capturing fine details and interpretable model structure while ensuring strong generalizability.

**Limitations.** Our multi-modal, multi-step pipeline focuses on discovering the data's structures, therefore current pipeline focuses on 1D datasets, and the pipeline's performance may depend on the quality of input visualizations. So the future work should include extending this to multivariate data to capture relationships between variables, and searching for the good quality of input visualizations that fits to VLM.

# 6 Acknowledgement

This work was supported by Samsung Electronics Co., Ltd (Project Code: IO240508-09825-01). T.-H. Oh was partially supported (30%) by Institute of Information & communications Technology Planning & Evaluation (IITP) grant funded by the Korea government(MSIT) (No. 2022-0-00124, No.RS-2022-II220124, Development of Artificial Intelligence Technology for Self-Improving Competency-Aware Learning Capabilities). T.-H. Oh was partially supported (30%) by Institute of Information & communications Technology Planning & Evaluation (IITP) grant funded by the Korea government(MSIT) (No. RS-2024-00457882, National AI Research Lab Project). T-H. Oh work was supported (40%) by Institute of Information & communications Technology Planning & Evaluation (IITP) grant funded by the Korea government(MSIT) (No.RS-2025-25443318, Physically-grounded Intelligence: A Dual Competency Approach to Embodied AGI through Constructing and Reasoning in the Real World). T.-H. Oh was partially supported by the KAIST Cross-Generation Collaborative Lab Project.

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

# A    Technical Appendices and Supplementary Material

## A.1    Gaussian Process Kernel Composition

In our paper, *model* means the *kernel composition* of gaussian process. Our kernel composition is done through below grammars and basis kernels. The basis kernels contain the linear(LIN), periodic(PER), squared exponential(SE), constant(C), white noise(WN) following [33]. And our composition grammar $\mathcal{O}$ contains addition(+), multiplication(x), replacement. So our model search space $\Sigma$ can be defined as:

$$\Sigma = \bigcup_{n=1}^{\infty} \{k \mid k \in \mathcal{B}, k = \oplus_{i=1}^{n}(b_i), b_i \in \mathcal{B}, \oplus_i \in \mathcal{O}\}, \tag{4}$$

$$\mathcal{B} ::= \text{Linear}|\text{Periodic}|\text{SE}|\text{WN}|\text{C} \tag{5}$$

$$\mathcal{O} ::= +| \times |\text{replacement} \tag{6}$$

where $\mathcal{B}$ represents the set of basis kernels and $\mathcal{O}$ denotes the set of kernel operations. A key property of this construction is that the space $\Sigma$ is closed under the specified operations , ensuring that any combination of basis kernels is valid and also belongs to the search space.

## A.2    Derivation of Visual Information Criterion

The derivation of visual criterion starts from the posterior probability of a model $\mathcal{M}$ given data $\mathcal{D}$:

$$p(\mathcal{M}|\mathcal{D}) = \frac{p(\mathcal{D}|\mathcal{M})p(\mathcal{M})}{p(\mathcal{D})}. \tag{7}$$

Typically, $p(\mathcal{M})$ is assume to be uniform, leading to the common Bayesian Information Criterion (BIC), which approximates $p(\mathcal{D}|\mathcal{M})$ using Laplace's approximation:

$$\text{BIC} = -2\log p(\mathcal{D}|\mathcal{M}, \theta^*) + k\log n, \tag{8}$$

where $\theta^*$ is the maximum likelihood estimator (MLE), $k$ is the number of model parameters, and $n$ is the sample size.

In this work, we modify the BIC by introducing:

1. A non-uniform prior $p(\mathcal{M}) \propto s_s(\mathcal{M})^{\alpha_s}$, where $s_s$ represents structure similarity score of the model $\mathcal{M}$, and $\alpha_s$ is a scale hyperparameter.

2. A modified likelihood function:
$$\tilde{p}(\mathcal{D}|\mathcal{M}, \theta) = s_f(\mathcal{D}|\mathcal{M}, \theta)^{\alpha_f} p(\mathcal{D}|\mathcal{M}, \theta), \tag{9}$$
where $s_f$ represents fitness score of the model $\mathcal{M}$ for data $\mathcal{D}$, and $\alpha_s$ is a scale hyperparameter.

The marginal likelihood is given by:

$$p(\mathcal{D}|\mathcal{M}) = \int \tilde{p}(\mathcal{D}|\mathcal{M}, \theta)p(\theta|\mathcal{M})d\theta. \tag{10}$$

As the fitness score is also maximized at the MLE $\theta^* = \arg\max p(\mathcal{D}|\mathcal{M}, \theta)$, one can apply Laplace's approximation around the MLE:

$$p(\mathcal{D}|\mathcal{M}) \approx \tilde{p}(\mathcal{D}|\mathcal{M}, \theta^*)(2\pi)^{k/2}|H|^{-1/2}, \tag{11}$$

where $H$ is the Hessian matrix of $-\log \tilde{p}(\mathcal{D}|\theta, M)$ evaluated at $\theta^*$:

$$H = -\frac{\partial^2}{\partial\theta^2}\log \tilde{p}(\mathcal{D}|\mathcal{M}, \theta)\bigg|_{\theta=\theta^*}. \tag{12}$$

Substituting into our approximation and taking the log on the both sides:

$$\log p(\mathcal{D}|\mathcal{M}) \approx \log \tilde{p}(\mathcal{D}|\mathcal{M}, \theta^*) - \frac{k}{2}\log n. \tag{13}$$

Then, our log posterior $\log p(\mathcal{M}|\mathcal{D})$ becomes:

$$\log p(\mathcal{M}|\mathcal{D}) \approx \alpha_f \log s_f(\mathcal{D}|\mathcal{M}, \theta^*) + \log p(\mathcal{D}|\mathcal{M}, \theta^*) - \frac{k}{2}\log n + \alpha_s \log s_s(\mathcal{M}) + C, \tag{14}$$

for some constant $C$.

Multiplying by 2 and rearranging gives:

$$2\log p(\mathcal{M}|\mathcal{D}) \approx -\text{BIC} + 2\alpha_f \log s_f(\mathcal{D}|\mathcal{M}, \theta^*) + 2\alpha_s \log s_s(\mathcal{M}) + C. \tag{15}$$

By modeling $2\log s_f(\mathcal{D}|\mathcal{M}, \theta^*) + 2\log s_s(\mathcal{M})$ with our EvaluatorVLM, and simply setting $\alpha = \alpha_s = \alpha_f$:

$$\text{VIC} = \alpha \cdot \text{EvaluatorVLM}(\mathcal{M}, \theta^*, \mathcal{D}) - \text{BIC}. \tag{16}$$

### A.3 Experimental Details

Our experiments are upon GPy and GPy-ABCD [56]. We have conducted each experiments for 5 rounds, with 10 random restarts, and used L-BFGS-B optimization. Also, we conduct top-3 sampling from model pool for each round. We have used gpt-4o-mini (for the main result) for both AnalyzerVLM and EvaluatorVLM, and we have set hyperparameter $\alpha$ to 50 of our EvaluatorVLM to balance with the BIC of the visual criterion. Also we have utilized the current round term for scoring to select mostly on recent models from the model pool. For Symbolic Regression, we have followed [48] and utilized its dataset. We have conducted each experiments for 20 rounds, with 5 random restarts each, and used scipy's optimize curve fit for parameter optimization. The function evaluation is done similarly to the gaussian process kernel discovery, setting the hyperparameter $\alpha$ to 0.05. To effectively search for the parameter for kernel search, we initially performed 10 random restarts to explore the parameter space broadly. Then we substituted the resulting parameters with those proposed by AnalyzerVLM. We then conducted a second-stage local optimization, using the AnalyzerVLM-initialized parameters as starting points. Our experiments are conducted on CPU with 16 cores for the precise calculation; and it may take around multiple hours for the experiments. The experiment for Gaussian Process Kernel Discovery, we have used the dataset of gpss-research, spliting training data into 9:1 for the validation data.

For BoxLM implementation, we have followed the explanation of [30]. For the fair comparison with our methods, we have set the basis kernel as linear(LIN), squared exponential(SE), constant(C), and white noise(WN), except the rational quadratic kernel(RQ), and also sampled top-3 models. Following our pipeline's evaluation, we have used Bayesian Information Criterion for the BoxLM's top-k model selection. For automatic statistician experiment, we have changed GPy-ABCD to work as greedy search through top-1 selection for each round. For ARIMA implementation, we have set ARIMA's p=2, d=1, q=2, and for facebook prophet implementation, we have set seasonality mode to multiplicative, and set changepoint prior scale to 0.1.

### A.4 More Qualitative Results

We present additional quantitative results in Fig. A12. Our method demonstrates superior performance overall capturing their underlying patterns. As shown, our method not only fits the training data well but also generalizes better mostly compared to other methods.

### A.5 Human Evaluation of the Model Selection

For human evaluation, the instructions are shown at Table A5, and example on Fig. A13. We have instructed the human evaluators to evaluate given model following EvaluatorVLM's criteria: visual fitness, and visual generalizability. For each criteria, we instructed evaluators to rate the given model from 1 to 5, giving the 5 for the best. For fitness, we have evaluated the mean prediction's fitness, and also the uncertainty region. So with small uncertainty region with good fitness, evaluators are instructed to give high score for visual fitness.

### A.6 Function Composition and Implementation Details for Symbolic Regression

In our experiment for symbolic regression, we have defined our basis function and base grammars and experiments for function composition following [39]. We have conducted the iteration for 20 rounds, and the parameter optimization is done through Scipy's optimize curve fit. For EvaluatorVLM, we have utilized $\alpha$ to 0.05, which can balance the original symbolic regression score function and the visual evaluation. We prompted AnalyzerVLM to propose the function composition based on the below basis functions and grammar, so our model search space of symbolic regression $\Sigma$ is:

$$\Sigma = \bigcup_{n=1}^{\infty} \{f \mid f \in \mathcal{B}, f = \oplus_{i=1}^{n}(b_i), b_i \in \mathcal{B}, \oplus_i \in \mathcal{O}\}, \tag{17}$$

$$\mathcal{B} ::= x|\sin(x)|\cos(x)|\tan(x)|\sinh(x)|\cosh(x) \tag{18}$$

$$\mathcal{O} ::= +| \times |sqrt| \exp | \log |abs \tag{19}$$

### A.7 Symbolic Regression Results at Real-World Dataset

We report the result of symbolic regression to the real-world univariate datasets [33], including Airline Passenger, Solar Irradiance, Mauna Loa, Wheat, Call-Center, Radio, and Gas Production. For the experiments, we conducted 20 rounds for each method. As shown in Table A6, SR-based methods require a large number of trials to identify appropriate models. Unlike AnalyzerVLM which proposes functions based on a detailed analysis of the data, symbolic regression-based methods like SGA [36] and LLM-SR [51] generate naive proposals without such insight, often leading to ineffective results. Although ICSR [39] enhances function proposal by utilizing visualization of data, enabling it to show better results SGA and LLM-SR, it still falls short of achieving the same level of performance as ours, due to the absence of precise, data-driven analysis. Also, symbolic regression-based

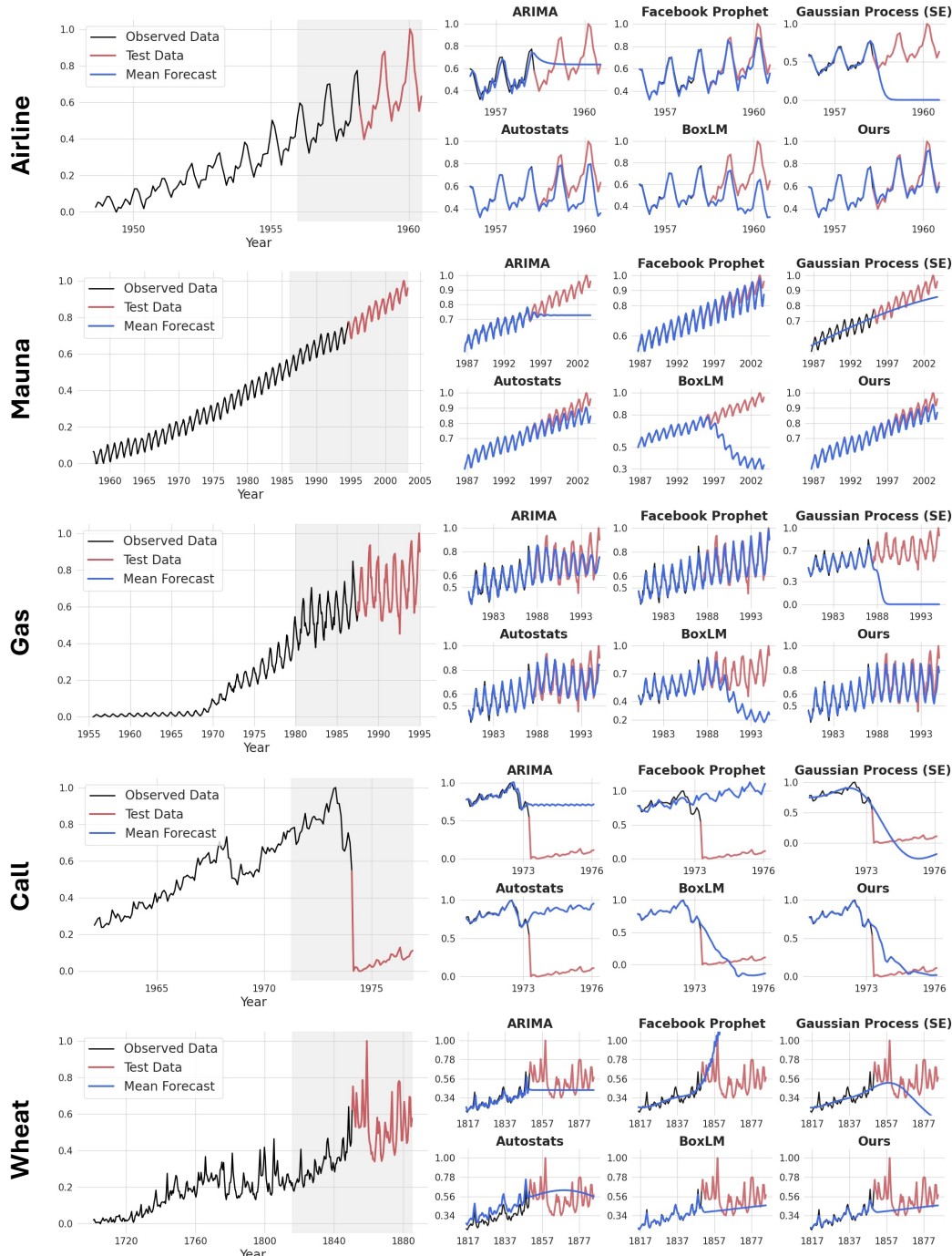

Figure A12: **More qualitative results.** We show more qualitative results at Airline, Mauna, Gas, Call, and Wheat data.

Please evaluate each model's fitness score. (You can give the same score to different *
models)

model 1                    model 3

model 2                    model 4

|         | 1 (worst) | 2 | 3 | 4 | 5 (best) |
|---------|-----------|---|---|---|----------|
| Model 1 | ○ | ○ | ○ | ○ | ○ |
| Model 2 | ○ | ○ | ○ | ○ | ○ |
| Model 3 | ○ | ○ | ○ | ○ | ○ |
| Model 4 | ○ | ○ | ○ | ○ | ○ |

Please evaluate each model's fitness score. (You can give the same score to different *
models)

model 1                    model 3

model 2                    model 4

|         | 1 (worst) | 2 | 3 | 4 | 5 (best) |
|---------|-----------|---|---|---|----------|
| Model 1 | ○ | ○ | ○ | ○ | ○ |
| Model 2 | ○ | ○ | ○ | ○ | ○ |
| Model 3 | ○ | ○ | ○ | ○ | ○ |
| Model 4 | ○ | ○ | ○ | ○ | ○ |

Figure A13: **Example screenshot of visual fitness for human evaluation.** The human evaluators are instructed to rate the given model from 1 to 5, having a same condition with the EvaluatorVLM.

Table A5: **Human Evaluation Instruction.** We have done the human evaluation similar to the EvaluatorVLM's evaluation instructions. Human evaluators are instructed to rate the visual fitness, and visual generalizability(e.g., consistency).

---

Evaluate the given model prediction following below criteria. Graph has three values: data(black line), mean prediction of the model(red line), and confidence region(blue shaded area).

**Fitness**: Compare the data plot(black line) and the model's mean(red line). If they are similar, give 5. If they are not similar at all(flatten line), give 1.
5: Red line perfectly matches black line
4: Red line closely follows black line with most details
3: Red line roughly follows black line (less detail, but not flat)
2: Red line is mostly flat or linear but somewhat follows data
1: Red line is flat and does not match data at all

**Uncertainty**: Check the blue shaded area is big. If blue shaded region is very small(means small uncertainty), give 5. If it is moderate, please give them 3. If it is very big(means high uncertainty) give them 1.
5: Confidence region almost invisible (very certain)
4: Confidence region very small
3: Confidence region small in the middle, larger at edges
2: Confidence region visible and consistent throughout
1: Confidence region very large everywhere (high uncertainty)

**Generalizability**: At each side, the model prediction may fail(showing flatten or sudden dropping prediction at both side, showing the large uncertainty region at both side), which means the generalization has failed. In this case, give generalibility 1. If you think the model prediction maintains at both side, give high score. 5: Predictions at edges are natural with small uncertainty
4: Predictions at edges are natural (not flat) with visible uncertainty
3: Predictions at edges hold but become mostly flat lines
2: Predictions at edges degrade noticeably
1: Predictions at edges fail badly, flattening or dropping sharply with large uncertainty

---

methods underperform in real-world datasets since they do not explicitly model the observation noise, while Gaussian process regression does.

Interestingly, we observe that symbolic regression's function composition can be a good starting point for the Gaussian process kernel discovery. To leverage this, we introduce a hybrid model discovery framework, denoted as Ours (SR + GP) in Table A6. We utilized simple SR framework [12] to generate the initial function composition and apply the top-3 function compositions' corresponding kernel structure with its initial parameters. Then, we conducted our GP kernel discovery pipeline from the kernels for 2 rounds. With this, our hybrid model discovery results in superior performance across all evaluated datasets. Moreover, it highlights the potential synergy between symbolic model discovery and probabilistic modeling for interpretable and accurate forecasting. Fig. A14 shows the qualitative results. As shown, utilizing the hybrid model discovery framework can enhance the performance and find the appropriate model across the dataset.

Table A6: **Quantitative results in symbolic regression.** We compare our pipeline with competing methods on the train and test region, reporting RMSE as the evaluation metric. On average, our pipeline achieves superior performance across datasets. **Bold** stands for the best, and underline for the second best.

| Method | Dataset | | | | | | | | | | | | | | Avg. | |
| | Airline | | Solar | | Mauna | | Wheat | | Call | | Radio | | Gas | | | |
| | Train | Test | Train | Test | Train | Test | Train | Test | Train | Test | Train | Test | Train | Test | Train | Test |
|---|---|---|---|---|---|---|---|---|---|---|---|---|---|---|---|---|
| SGA [36] | 0.0700 | 0.1668 | 0.1451 | 0.2652 | 0.0332 | 0.0354 | 0.0599 | **0.1338** | 0.0689 | 0.8583 | 0.2859 | 0.8025 | 0.0574 | 0.3424 | 0.1029 | 0.3720 |
| LLM-SR [51] | 0.0692 | 0.3304 | 0.1039 | 1.7142 | 0.0317 | 0.2096 | **0.0492** | 0.1438 | 0.0438 | 21.028 | 0.1875 | 0.1798 | 0.0490 | 0.6521 | 0.0763 | 3.4659 |
| ICSR [39] | 0.0420 | 0.1029 | 0.1807 | 0.3845 | 0.0347 | 0.0343 | 0.0495 | 1.4430 | 0.0548 | 0.4725 | 0.1799 | 0.2427 | 0.0497 | 0.1672 | 0.0844 | 0.4067 |
| **Ours (SR + GP)** | **0.0194** | **0.0354** | **0.0297** | **0.3345** | **0.0037** | **0.0166** | 0.0150 | 0.1801 | **0.0095** | **0.1088** | **0.0491** | **0.0515** | **0.0159** | **0.0581** | **0.0203** | **0.1121** |

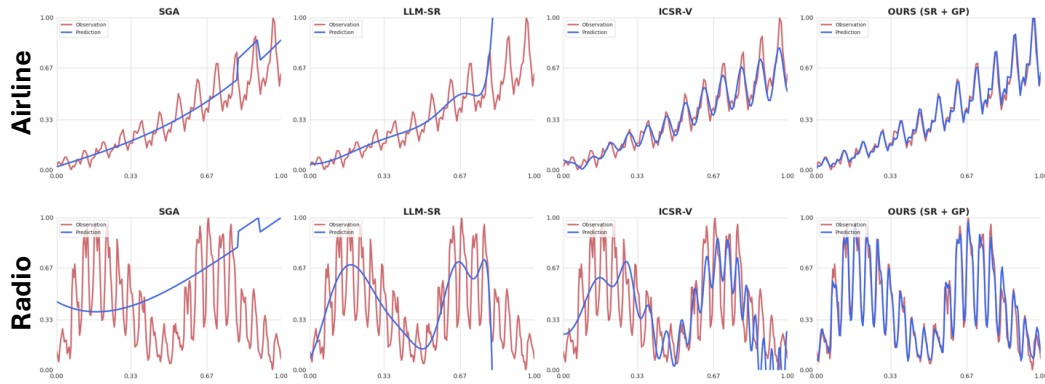

Figure A14: **Qualitative result in symbolic regression.** Although conventional symbolic regression methods often exhibit limited performance on real-world datasets, our hybrid model discovery framework demonstrates notable improvements.

## A.8 AnalyzerVLM Prompts

For the AnalyzrVLM's System Prompt, we have utilized WaitGPT [64]'s system prompt. Amd VLM's action choosing(*e.g.* analysis, code generation, model proposal) prompt is shown at A7. As shown here, we do not simply give AnalyzerVLM the data itself, instead, we provide AnalyzerVLM a small code block that can acess the data and the model's predictions. With this, AnalyzerVLM can inject this code block at the code generating action, and get the executed output of the code. In this way, AnalyzerVLM can choose how to represent the data and model, not just fully stuffing the long sequence of the numbers into itself. We also report the prompts used for AnalyzerVLM and EvaluatorVLM for symbolic regression experiments at Table A9. Our implementation of prompts follows the scheme of [48], utilizing our multi-step reasoning for AnalyzerVLM and visual evaluation for EvaluatorVLM.

## A.9 Prompt for AnalyzerVLM and EvaluatorVLM at Symbolic Regression

We report the prompts used for AnalyzerVLM and EvaluatorVLM for symbolic regression experiments at Table A9. Our implementation of prompts follows the scheme of [48], utilizing our multi-step reasoning for AnalyzerVLM and visual evaluation for EvaluatorVLM.

## A.10 EvaluatorVLM Prompts

EvaluatorVLM's evaluation prompts are shown at Table A8. We evaluate fitness and structure similarity seperately, and add two scores and use for the final score. For consistency, we evaluate each parts(*e.g.* fitness, structure similarity) twice, and averaged.

First, we evaluate the fitness based on two points: mean prediction's data sample matching, size of confidence area. For the mean prediction and data matching, we prompted VLM to score 0-10 points if the mean prediction is just constant value, 10-20 if it is linear but follows overall trend, upper if the mean prediction follows more trend than that. For the confidence area, we gave 30-50 score if confidence area is small, and gave under 30 if confidence area gets sudden big at each side, which means it looses the confidence at extrapolation region.

Second, we evaluate the structure similarity based on the red line's structure similarity at the middle of the graph (train region) is maintained at the ends of graph. We prompted VLM to give high score(40-50) when the structure similarity is maintained, else give the lower score.

## A.11 Detailed Multi-step Analysis Example

And we also show the detailed outputs of multi-step analysis of AnalyzerVLM, at Fig. A15, Fig. A16, Fig. A17, and we can see that our AnalyzerVLM chooses how to start the analysis, first, our AnalyzerVLM plans to first visualize data and the predictions, and generates the code for it. When the code is executed and the output $O$ (e.g., visualization plot, the covariance visualization) is given, AnalyerVLM repeats the its process until the analysis is sufficient.

Table A7: **AnalyzerVLM Prompts.** We provide the prompt for analyzerVLM. First it describes about the base kernel, and provide the actions that AnalyzerVLM can select. Then AnalyzerVLM is prompted to select the actions.

---

**AnalyzerVLM: Analysis and action choosing.**

---

Task Overview: You are provided with the mean and covariance 1D array of the fitted kernel [`'kernel'`]. Your job is to either:
Generate Python code for further analysis, or Recommend new kernel combinations.
You can only choose one action at a time.
Kernel Adjustment Options:
You can adjust the current kernel by forming new combinations with base kernels using the following operations:
- Addition (S + B): Add a new base kernel B to the current kernel S.
- Multiplication (S * B): Multiply the current kernel - - S with a new base kernel B.
- Base Kernel Replacement: Replace the base kernel - B with a new base kernel B'.

Base Kernels Available:
- Linear (LIN)
- Periodic (PER)
- Squared Exponential (SE)
- Constant (C)
- White Noise (WN)

Action 1: Analyze the Fitted Kernel (Python Code)
If you need further analysis before making a recommendation, generate Python code for the task. You can draw insights from the mean, covariance, and confidence intervals of the fitted kernel, or analyze the parameter itself. However, please try one analysis at a time.
Access the Data and the Model Parameters:

```python
X, y, enX, en_mean, en_cov, en_low_quantile, en_high_quantile = access_data
(fitted_models[0])
model_printout(model)
```

This will give you train data $(X, y)$, enlarged data with test data (`enX`), the mean, covariance, and confidence intervals for the enlarged `X`, and the model parameters.
Generate Code: If analysis is needed, provide the Python code necessary to calculate or visualize key insights.

```python
Python code goes here
```

Action 2: Recommend Kernel Combinations
If you have already analyzed the kernel, suggest new kernel combinations using the current kernel S and the base kernels. Use the operations outlined above.
Example Recommendations:

```
next kernels: ["new combination1", "new combination2", "new combination3",
"new combination4", "new combination5", "new combination6"...]
```

Important: Choose only one action: Either provide Python code or recommend new kernel combinations.
Do not provide both at the same time.

---

Table A8: **EvaluatorVLM prompt.** For the fitness evaluation, we have evaluated how well the real data an mean prediction fits, and how small the confidence area is. Each points are measured at 50 points, total to 100. For the generalizability evaluation, we have evaluated how the structure is maintained throughout the data. Structure similarity is measured out of 50 points

---

**EvaluatorVLM: Fitness evaluation of visual information criterion.**

---

You will evaluate the similarity of the two graph, data graph and predicted mean graph. Assign a score from 0 to 50. Evaluate the Structure Similarity Between Real Data and Mean Prediction.

Please check the real data graph is similar to predicted mean graph. Please check below:
- Mean graph is similar with sample graph (20-50 points).
- Predicted mean graph is linear line while it shares trend with data graph (10-20 points)
- Mean graph is linear and it does not share the trend at all(0-10 points).
Please generate the response in the form of a Python dictionary string with keys of kernel name. score is in INTEGER, not STRING.
Please evaluate how similar the two graphs are. First is data graph and second graph is predicted mean graph. Output should be the score for the kernel1. kernel1:

Please evaluate how small the confidence interval area is.
Evaluate the Size of the Confidence Area (LightBlue Shaded Area)
- Confidence scores should be assigned based on the size of the lightblue shaded area. So do not consider the red line and black line, only the lightblue shaded area's size and the region of uncertainty.
Please check what the confidence area looks like. Assign a score from 0 to 50 following below:
1. Confidence interval area is hard to see, uncertainty is small(this case assign 40-50 points).
2. Confidence interval area is hard to see in the middle of graph, but large at the boundaries (30-40 points). This means the model is overfitted to the middle, so give a low score.
3. Confidence interval area is normal in the middle, uncertainty remains but acceptable or becomes larger over y at the boundaries (0-30 points).

---

**EvaluatorVLM: Generalizability evaluation of visual information criterion.**

---

You will evaluate how well the predicted kernel (red line) maintains based on the below criteria:
Evaluate the structure similarity of middle of the graph and the ends of the graph.
Check the blue line's structure similarity of the middle maintains at the left and right end of the graph. If it was following the data well but suddenly changes to the constant line at the ends of the graph, assign low score for structure similarity score. But if structure similarity is maintained, assign 40-50 score.
Please generate the response in the form of a Python dictionary string with keys of kernel name. score is in INTEGER, not STRING.
Please evaluate how similar the two graphs are. First is data graph and second graph is predicted mean graph. Output should be the score for the kernel1. kernel1:,

---

Table A9: **AnalyzerVLM and EvaluatorVLM prompts for symbolic regression.** We report action choosing prompt used by AnalyzerVLM and fitness & generalizability evaluation prompt employed by EvaluatorVLM in symbolic regression.

---

**AnalyzerVLM: Action choosing prompt.**

---

Your task is to give me a list of five new potential functions that are different from all the ones reported below, and have a lower error value than all of the ones below. Before the function generation, please first analyze the given data points and reported functions first(e.g., visualization, or get the statistics). Guess and list up which this function would be. If you generate the Python code that includes your analysis, I will execute and give you the result. You can use sympy for checking the function prediction. For saving the visualization, please avoid using plt.show(), and use plt.savefig('./ztmpimgs/imagename') when imagename is any visualization you made. Before using the data points, please sort them first.
Please give me only python code for now. Code:

```python
Python code goes here
```

Please try to bulid upon the function with the smallest error, then generating different ones too. Generate as diverse as diverse functions!

---

**EvaluatorVLM: Fitness evaluation prompt.**

---

You are an intelligent chatbot designed for evaluating two graph's similarity.
You will evaluate the structure similarity of the two graph, data graph and predicted mean graph. Assign a score from 0 to 50.
Evaluate the Structure Similarity Between Real Data and Mean Prediction.
Please check the real data graph is similar to predicted mean graph. Please check below:
- Mean graph is similar with sample graph (20-50 points).
- Predicted mean graph is linear line while it shares trend with data graph (10-20 points)
- Mean graph is linear and it does not share the trend at all(0-10 points).
Please evaluate how similar the two graphs are. First is data's line plot, and second is predicted value's line plot. Please evaluate how well the predicted value fits to the data. Output should be the score for the function1. Please generate the response in the form of a Python dictionary string with keys of function name. score is in INTEGER, not STRING.
function1:

---

**EvaluatorVLM: Generalizability evaluation prompt.**

---

You are an intelligent chatbot designed for evaluating the correctness of each functions.
You will evaluate how well the predicted value (red line) fits based on the below criteria:
- Evaluate the structure similarity of middle of the graph and the ends of the graph.
- Check the blue line's structure similarity of the middle maintains at the left and right end of the graph.
- If it was following the data well but suddenly changes to the constant line at the ends of the graph, assign low score for structure similarity score.
But if structure similarity is maintained, assign 40-50 score.
Please generate the response in the form of a Python dictionary string with keys of function name. 'score for structure similarity' are in INTEGER, not STRING.

---

Let's start with **Action 1: Analyze the Fitted Kernel** by generating Python code for further analysis.

The goal is to visualize the mean, covariance, and confidence intervals of the fitted kernel.

Plan:
1. Access the data and model parameters using the provided function.
2. Visualize the data, mean, and confidence intervals using Matplotlib.

Here's the Python code for the analysis:
PYTHON CODE

Recap of the Plan:
1. Access the data and model parameters.
2. Visualize the data, mean, and confidence intervals using Matplotlib.

Now, you can run this code to perform the analysis. Let me know if you need any further assistance!

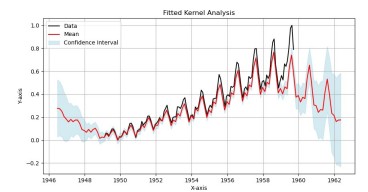

fitted_kernel_analysis.png

Analysis of the Current Kernel (LIN + SE * (PER + C))

Based on the provided graph and the output from the previous analysis, here are the key observations:

1. **Mean Value vs. Real Data Points**:
   - The mean (red line) closely follows the actual data points (black line) for most of the time, indicating that the current kernel captures the overall trend well.
   - However, there are periods where the mean deviates from the actual data, particularly towards the end of the time series. This suggests that the model may not be fully capturing the underlying dynamics in those regions.

2. **Covariance**:
   - The confidence intervals (light blue shaded area) are relatively wide, especially at the boundaries. This indicates a higher uncertainty in the predictions, which could be due to the non-stationary nature of the data or the choice of kernels.
   - A smaller covariance would generally indicate more confidence in the predictions, so this is an area for potential improvement.

3. **Parameter Appropriateness**:
   - The current kernel effectively captures both linear and periodic trends, but the presence of the constant (C) and the squared exponential (SE) may not be necessary if the data is primarily linear with periodic fluctuations.
   - The periodic component (PER) is evident in the data, but the SE kernel may be introducing unnecessary complexity.

Next Steps
Given the analysis, it seems that further analysis is warranted to **explore the residuals of the model**, which can provide insights into how well the current kernel is performing and where it may be lacking. This will help in making a more informed recommendation for new kernel combinations.

Python Code for Residual Analysis
Here's the code to calculate and visualize the residuals:

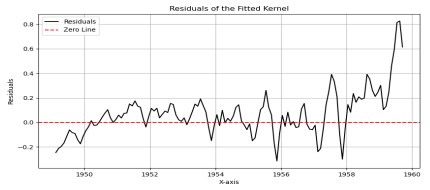

residuals_analysis.png

Figure A15: **Example of VLM Analysis - step 1,2.**

## Analysis of the Residuals

Based on the residuals plot generated from the previous analysis, here are the key observations:

1. **Residuals Behavior**:
   - The residuals (black line) fluctuate around the zero line (red dashed line), which is a good sign as it indicates that the model does not consistently overestimate or underestimate the actual values.
   - However, there are periods where the residuals show significant deviations from zero, particularly towards the end of the time series. This suggests that the model may not be capturing some underlying patterns or trends effectively in those regions.

2. **Variance of Residuals**:
   - The variance of the residuals appears to increase over time, especially in the later years. This indicates that the model's uncertainty is growing, which could be a sign that the current kernel is not adequately capturing the dynamics of the data.

3. **Model Appropriateness**:
   - The current kernel (LIN + SE * (PER + C)) seems to capture the overall trend well, but the increasing variance in the residuals suggests that the model may be too simplistic or not flexible enough to account for the complexities in the data.
   - The presence of the constant (C) may not be necessary, as the data appears to have a more complex structure that could be better captured by other kernels.

### Next Steps
Given the analysis of the residuals, it seems that further analysis is warranted to explore the trends in the residuals, **such as calculating the autocorrelation of the residuals.**
This can help identify if there are any patterns in the residuals that the current model is failing to capture.

### Python Code for Autocorrelation Analysis
Here's the code to calculate and visualize the autocorrelation of the residuals:

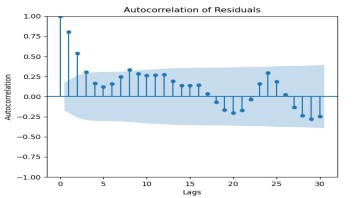

residual_autocorrelation_analysis.png

## Analysis of the Autocorrelation of Residuals

Based on the autocorrelation plot of the residuals, here are the key observations:

1. **Autocorrelation Behavior**:
   - The autocorrelation values are significant for the first few lags, indicating that there is a correlation between the residuals at these lags. This suggests that the model may not be fully capturing the temporal dependencies in the data.
   - The autocorrelation drops off quickly after the first few lags, but the presence of significant values indicates that the residuals are not independent, which is a violation of the assumptions of many statistical models.

2. **Implications for the Current Kernel**:
   - The presence of autocorrelation in the residuals suggests that the current kernel (LIN + SE * (PER + C)) may not be flexible enough to capture the underlying structure of the data. This could lead to systematic errors in predictions.
   - The model may benefit from incorporating additional kernels that can better account for the temporal dependencies observed in the residuals.

3. **Variance of Residuals**:
   - The residuals show some variability, but the autocorrelation indicates that there are patterns that the current kernel is not capturing. This could lead to increased uncertainty in predictions, especially in the later periods.

### Next Steps
Given the analysis of the autocorrelation of the residuals, it seems that further analysis is warranted to explore the trends in the data itself, such as calculating the period of the periodic component. This can help in making a more informed recommendation for new kernel combinations.

### Python Code for Period Calculation
Here's the code to calculate and visualize the period of the data:

CODE EXECUTION ERROR OCCURED

Figure A16: **Example of VLM Analysis - step 3,4.**

Analysis of the Current Kernel (LIN + SE * (PER + C))

Based on the previous analyses, here are the key observations regarding the current kernel:

1. **Mean Value vs. Real Data Points**:
   - The mean value generated by the kernel follows the real data points reasonably well, but there are noticeable deviations, especially towards the end of the time series. This indicates that the kernel may not be capturing all the underlying trends effectively.

2. **Covariance**:
   - The residuals showed significant autocorrelation, indicating that the model's predictions are not independent. This suggests that the covariance structure of the current kernel may not be appropriate for the data, leading to increased uncertainty in predictions.

3. **Parameter Appropriateness**:
   - The current kernel captures both linear and periodic trends, but the presence of the constant (C) and the squared exponential (SE) may not be necessary. The increasing variance in the residuals suggests that the model may be too simplistic or not flexible enough to account for the complexities in the data.

Next Steps
Given the analysis, it seems that further analysis is warranted to explore the periodicity of the data more robustly. **Since the previous attempt to calculate the period resulted in an error, I will provide a different approach to analyze the periodicity.**

Python Code for Periodicity Analysis
Here's an alternative code to analyze the periodicity using Fourier Transform, which can help identify dominant frequencies in the data:

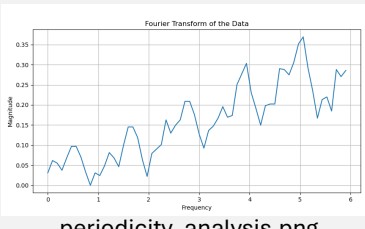

periodicity_analysis.png

Analysis of the Periodicity Analysis Graph

Based on the Fourier Transform plot of the data, here are the key observations:

1. **Frequency Components**:
   - The plot shows several peaks at different frequencies, indicating that the data has multiple periodic components. The presence of these peaks suggests that the data exhibits periodic behavior, which the current kernel may not fully capture.

2. **Magnitude of Frequencies**:
   - The magnitudes of the frequencies indicate the strength of the periodic components. The highest peaks suggest dominant frequencies that could be important for modeling the data. This information can be useful for selecting appropriate kernels that can capture these periodic trends.

3. **Implications for the Current Kernel**:
   - The current kernel (LIN + SE * (PER + C)) **may not be adequately capturing the periodic nature of the data**, especially given the multiple frequency components observed. The linear and squared exponential components may not be sufficient to model the complexities in the data.

Next Steps
Given the analysis of the periodicity, it seems that further analysis is warranted to explore the specific periodic components and how they can be incorporated into the kernel. However, since we have already conducted several analyses, it may be more beneficial to recommend new kernel combinations based on the insights gained.

Recommended Kernel Combinations
**Considering the periodic nature of the data and the limitations of the current kernel**, here are some new kernel combinations that could be explored:

```plaintext
next kernels: ["PER * (LIN + SE)", "PER + (LIN * SE)", "PER * SE + LIN", "LIN + PER * C", "SE * PER + C", "PER * (LIN + C)"]
```

Figure A17: **Example of VLM Analysis - step 5,6.**

