# OpenReview forum: "Automated Model Discovery via Multi-modal & Multi-step Pipeline"
_NeurIPS.cc/2025/Conference — NeurIPS 2025 poster_

### Official Review · Reviewer_MXP2 · 2025-07-03

**Clarity:** 2
**Significance:** 3
**Originality:** 3
**Rating:** 4
**Confidence:** 3

**Summary:**

This work focuses on automated model discovery in 1D data. A multi-step pipeline is designed and the pipeline leverages the multimodal information. An interesting part is replacing the role of human in the pipeline in observing results figures with a vision-language model to understand the results via capturing relationships and trends. The evaluation part also benefits from leveraging automatically generated code. Quantitative results are demonstrated to justify the effectiveness of the proposed approach in the discovered model’s performance in downstream datasets. Ablation study also demonstrate the performance of leveraging multimodal information compared to text only information.

**Questions:**

An example on the resulting model and corresponding parameters would facilitate the understanding.

What’s the resolution of the results plot? Any example on the code generated?

How exactly are the results in Figure 3 for LLM generated? Are text based results fed into the model in pure number series or are they incorporated into python code using matplotlib (or whatever other visualization tools) which are further used to generate visual figures? For a fair comparison I expect the text-form results to be generated from something in code-style.

Line177 typo.

**Ethical Concerns:**

["NO or VERY MINOR ethics concerns only"]

**Final Justification:**

The authors have addressed my concerns so my final rating is positive. Considering the work focuses only on 1D data, the impact might be somewhat limited so I don't rate a higher score.

**Limitations:**

yes.

**Paper Formatting Concerns:**

no.

**Quality:**

3

**Strengths And Weaknesses:**

Strengths:

This work explores the possibility of leveraging multimodal large language models to replace the role of human in automated model architecture design, which is interesting.
Experiments are conducted in multiple datasets with competitive results.
The multi-step pipeline design is reasonable in the form of an agent system.



Weaknesses:

The analysis is not comprehensive enough [Figure 2]: How the model processes different visualizations [line 47-48]? For example, human often observe the plots to identify whether a certain model is overfitting, is underfitting, or has a good generalization capability, or the loss might be strongly shaking, or observe some unnormal curves. How does the proposed approach perform in each case? The meaning of the y-axis in a plot could also significantly change the information a plot delivers (e.g., accuracy is increasing or the loss is increasing).

I think the visual quality is indeed important. This component is a clear difference compared to prior work but this component is not comprehensively analyzed regarding its quality, complexity, resolution, etc.

[line25-26] how does this approach balance interpretability and model fit?

What code is generated to evaluate what aspects? There are many aspects to evaluate including OOD, robustness, etc. How to automate these aspects via LLM? I understand it might be too much to be included in one single paper, but relevant discussions would be helpful.

[line39-40] what is the tool pool? What tools exist and what does not exist in the searching space?

I like the general idea but more evaluations and implementations details would strengthen the contributions and facilitate the understanding.

---

> ### Author Rebuttal · Authors · 2025-07-31
>
> We thank reviewer MXP2 for valuable comments, and recognizing our work as:
>
> - **Possibility of leveraging multimodal large language** models to replace the role of human **in automated model architecture design**, which is interesting
> - Replacing the role of human in the pipeline in observing results figures with a vision-language model as i**nteresting**, to understand the results via capturing relationships and trends
> - and also checking our typo
>
> We have addressed MXP2’s concerns and questions below.
>
> ### **Ablation study on different visualizations**
>
> > *How the model processes different visualizations \[line 47-48\]?*
>
> We agree with your comment that to interpret the plot correctly requires not only the visual structure of the visualization, but also the semantic meaning of its axis and different visual quality.
> As the reviewer suggested, we have also conducted how EvaluatorVLM responds to different axis plot visualizations (loss curves, accuracy) for three different types of cases: 1) model overfitting, 2) model underfitting (unstable training & validation curves), and 3) model well-fitted (stable training & validation curves).
> In the case, we have found that EvaluatorVLM has evaluated the underfitting or overfitting case with the given loss curve or accuracy, interpreting the case with stable training loss and unstable validation loss as a low generability case, and a relatively higher fitness score. With the shaking loss curve, EvaluatorVLM judged the model as low fitness and low generalizability, assigning low scores.
>
>
> > *I think the visual quality is a clear difference compared to prior work but this component is not comprehensively analyzed regarding its quality, complexity, resolution…*
>
> We also conducted an ablation study of the plot components to examine how EvaluatorVLM processes different plot quality, including resolution, color, and thickness. We report the correlation between human annotation and EvaluatorVLM’s answer. As shown, good results were obtained with 1) resolution 800x600, 600x250 2) red and blue line are preferred and 3) linewidths 1, 2 are most preferred. For fitness, thinner lines (1, 2) were preferred for fine-grained comparison, and for generalizability, linewidth 3 is preferred.
> We thank the reviewer for the important comments, and we will add the ablation study to the final version.
>
> | Resolution | 128x128 | 300x200 | 400x300 | 600x600 | 800x600 | 600x400 | 600x250 | 1200x500 |
> | :---- | :---- | :---- | :---- | :---- | :---- | :---- | :---- | :---- |
> | visual fitness | 0.293 | 0.635 | 0.595 | 0.410 | 0.612 | 0.647 | **0.725** | 0.595 |
> | visual generalibility | 0.194 | 0.630 | 0.601 | 0.586 | **0.844** | 0.658 | 0.699 | 0.511 |
> | average | 0.244 | 0.632 | 0.598 | 0.498 | **0.728** | 0.653 | 0.712 | 0.553 |
>
> | Color | Red | Blue | Green |
> | :---- | :---- | :---- | :---- |
> | visual fitness | **0.812** | 0.804 | 0.703 |
> | visual generalibility | 0.673 | **0.782** | 0.719 |
> | average | 0.743 | **0.793** | 0.711 |
>
> | Linewidth | linewidth=1 | linewidth=2 | linewidth=3 | linewidth=4 | linewidth=5 |
> | :---- | :---- | :---- | :---- | :---- | :---- |
> | visual fitness | 0.764 | **0.817** | 0.622 | 0.569 | 0.631 |
> | visual generalibility | 0.766 | 0.742 | **0.799** | 0.753 | 0.744 |
> | average | 0.765 | **0.780** | 0.711 | 0.661 | 0.688 |
>
> ### **Methods for balancing interpretability and model fit**
>
> Traditional methods for interpretable model selection have utilized a complexity term to regularize the model not to be in a complex form. However, such complexity-based regularization alone is often insufficient for discovering truly meaningful or high-quality models. Our method addresses this limitation by going beyond complexity control, incorporating human-like model suggestion and evaluation to better balance interpretability and model fit.
>
> ### **AnalyzerVLM’s generated code analysis**
>
> > What code is generated to evaluate what aspects? There are many aspects to evaluate including OOD, robustness, etc...*
>
> We have shown the example of AnalyzerVLM's analysis at the Appendix. AnalyzerVLM focuses on suggesting a good model with the analysis, covering OOD analysis and whether the model is robust. We categorize the behaviors of AnalyzerVLM as follows:
>
> 1\. Numerical analysis
>
> - Directly printing out the data or predictions sometimes
> - Period estimation: using scipy.signal.find\_peaks() to detect and calculate peak intervals
> - Linearity estimation: via np.polyfit() to evaluate global linear trends
>
> 2\. Plot-based visualization
>
> - When data & model access code snippet is given, AnalyzerVLM tends to choose visualizing data and model’s prediction using plt.plot(), analyzing the plot whether the overall trend suddenly drops or maintained at OOD regions.
> - AnalyzerVLM tends to calculate the residual of data and model’s prediction, to check whether current model is fully reflecting the data’s trend. If the residual shows any characteristic, AnalyzerVLM tends to assign a new model with its characteristic.
> - To check the residual’s characteristics, AnalyzerVLM tends to draw the residual distribution or histogram (to check whether it is skewed), or the correlations between input and residuals.
>
> We also provide the examples of AnalyzerVLM checking the data’s characteristics and residual distribution visualization below.
> ```python
> # Analyze for periodicity
> peaks, _ = find_peaks(y)
> periods = np.diff(X[peaks])
> if len(periods) > 0:
>     estimated_period = np.mean(periods)
> else:
>     estimated_period = None
>
> # Analyze for smoothness (lengthscale)
> differences = np.diff(y)
> lengthscale = np.mean(np.abs(differences)) if len(differences) > 0 else None
>
> # Analyze for linearity (slope and offset)
> slope, intercept = np.polyfit(X, y, 1)
> print(f"Estimated Period: {estimated_period}")
> print(f"Estimated Lengthscale: {lengthscale}")
> print(f"Estimated Slope: {slope}, Estimated Offset: {intercept}")
>
> # Visualize residuals distribution
> plt.figure(figsize=(10, 6))
> sns.histplot(residuals, bins=30, kde=True, color='blue')
> plt.axvline(0, color='red', linestyle='--', label='Zero Line')
> plt.savefig('./tmpimgs/residual_distribution_analysis.png')
> ```
>
>
> ### **What tools exist and what does not exist in the searching space?**
> For the tool pool, we did not explicitly give the list of tools that AnalyzerVLM can use, but relies on AnalyzerVLM’s internal ability to analyze using Python. But we have checked that AnalyzerVLM mostly relies on such tools: 1) matplotlib and seaborn, 2) numpy, 3) scikit-learn 4) scipy. With such tools, AnalyzerVLM conducts the multi-step analysis for recognizing its data’s characteristics.
>
> ### **Example of the resulting model and corresponding parameters**
>
> To demonstrate this, we provide the example of the searched kernel result of Radio dataset (Fig.2).
>
> Our discovered model PER * (PER + SE) revealed two periodic components: 10.25 years for period 1, 1.01 years for the second period, which aligns the data’s characteristics with the multiple periods that the dataset has. Also SE showed small local structure, smoothed trend (SE lengthscale: 53.8 years).
>
> Overall, this result shows that our pipeline can automatically discover models with interpretable kernel structures and parameters, revealing meaningful characteristics such as periodicities and smooth trends inherent in the data.
>
> ### **Clarification of Figure 2 & 3**
>
> > *What’s the resolution of the results plot?*
>
> The resolution of the results plot (Fig. 2) is 2000x560.
>
> > *How exactly are the results in Figure 3 for LLM generated? Are text based results fed into the model in pure number series or are they incorporated into python code using matplotlib...*
>
> We understand the concerns about the settings of text-based analysis and our multimodal analysis at Figure 3\. As you mentioned, for text-based results, we did not feed the pure number series but gave the code snippets for accessing data to be incorporated into python code. Also we restricted it not to use visualization tools, but allowed only numerical analysis results. We have seen that the AnalyzerLLM conducts numerical analysis to understand data’s overall structure: linearity check (to find out its shape is linear), or periodicity check (peak interval variance), or mean/variance check.

---

> > ### Comment · Reviewer_MXP2 · 2025-08-05
> >
> > The reply to "How the model evaluates plots/curves indicating under-fitting, overfitting, well-fitting and un-normal curves" is quite vague without concrete results.
> >
> > Overall I think my concerns are mostly resolved and tend to keep my positive score.

---

> ### Author Response · Authors · 2025-08-06
> **Response to Reviewer MXP2**
>
> In our understanding, reviewer MXP2 asked **EvaluatorVLM's behaviors when the model is in under-/well-/over-fitting cases**. If our understanding is wrong, please correct us. We are happy to discuss further.
>
> To provide additional evidence, we designed a controlled experiment using synthetic data generated from a 3rd-degree polynomial function: $−x^3+3x$.
> We then fitted polynomials of degrees 1 through 6 to **simulate under-fitted (degree 1-2), well-fitted (degree 3), and over-fitted (degree 4-6) cases**.
> For each model, we evaluated both fitness and generalizability scores based on the data and prediction plot (averaged over 5 evaluation per score).
>
> The results are summarized below (*: well-fitted):
>
> | degree  | 1   | 2   | 3*   | 4   | 5   | 6   |
> | ------- | --- | --- | --- | --- | --- | --- |
> | fitness | 15  | 15  | **45**  | 32  | 42  | 40  |
> | gen.   | 10  | 10  | **48**  | 30  | 40  | 10  |
>
> As shown, EvaluatorVLM has assigned both low fitness and generalizability score for the underfitted models (degree 1, 2). For a well-fitted model (degree 3), EvaluatorVLM has assigned both high fitness and generalizability score. For higher degrees (degree 4, 5, 6), EvaluatorVLM began to assign relatively lower scores, meaning that our EvaluatorVLM constraints the over-fitted models with low scores.
>
> These results show that EvaluatorVLM is capable of detecting **whether a model aligns well with the data** and whether it **fails to generalize due to underfitting or overfitting** like humans, and validating its role as a practical component in our visual model evaluation pipeline.

---

> ### Author Response · Authors · 2025-08-08
> **Further Discussion**
>
> We appreciate reviewer MXP2, for taking the time to review our submission and providing such constructive feedback.
>
> If there are any parts of our rebuttal that remain unaddressed, please feel free to request a discussion, we’d be happy to respond. Additionally, we’d be happy to answer any further questions you may have.
>
> Best regards,
>
> Authors

---

> > ### Comment · Reviewer_MXP2 · 2025-08-09
> >
> > Thanks the authors for the efforts in the rebuttal. I have no more concerns and would keep my positive rating. Considering this work mainly focuses on the scenario of model discovery in 1D data, the impact might be somewhat limited so I am not raising the score further. But I do see this direction as promising to be extended to higher dimensional data.

---

### Official Review · Reviewer_fF4a · 2025-07-03

**Clarity:** 3
**Significance:** 2
**Originality:** 3
**Rating:** 4
**Confidence:** 4

**Summary:**

This paper proposes a multi-modal and multi-step pipeline for automated model discovery. The core innovation lies in integrating vision-language models into the model proposal and evaluation process. Specifically, the authors introduce AnalyzerVLM, which conducts iterative analyses by dynamically generating and executing code to understand data trends and propose candidate models, and EvaluatorVLM, which assesses model quality using a novel Visual Information Criterion. VIC combines traditional Bayesian Information Criterion with VLM-based visual evaluations to balance model fit and generalizability. The pipeline operates in four stages: model proposal, parameter fitting, evaluation, and selection, iterating these steps to refine model quality.
The authors evaluate their approach on several real-world time series datasets and symbolic regression tasks, showing that it outperforms traditional methods and recent LLM-based systems in both training fit and test generalization. Ablation studies highlight the importance of multi-modal representations and multi-step reasoning for achieving strong performance. Overall, the work aims to mimic human expert behaviors in automated scientific discovery, reducing reliance on predefined grammars and manual interventions.

**Questions:**

1.The method is currently demonstrated only on univariate datasets. Do the authors have thoughts or preliminary experiments on how their approach could handle multivariate or high-dimensional data, especially given the challenges of visualizing such data?
2.Appendix A.5 mentions averaging VLM assessments across two runs, but how consistent are these evaluations? Could the authors provide quantitative evidence of agreement between VLM outputs, or discuss how the system handles conflicting evaluations?
3.In scientific applications of symbolic regression, model interpretability and formula readability are often as critical as predictive accuracy. Simple, human-readable expressions can provide insights and foster understanding of the underlying phenomena. Could the authors comment on the interpretability of the symbolic regression formulas discovered by their method? Are there mechanisms within the proposed pipeline—or could there be extensions—to encourage the discovery of more concise and interpretable expressions?

**Ethical Concerns:**

["NO or VERY MINOR ethics concerns only"]

**Final Justification:**

I appreciate the authors addressed most of my questions and argued about my concerns. But I still don't think it is reliable to use VLMs to judge fitness and structure similarity. Furthermore, the rebuttal approved the correctness of the paper, but didn't change the significance and novelty of the paper, so I keep my score.

**Limitations:**

Yes. The authors explicitly discuss the current restriction to 1D datasets and the reliance on visualization quality in Section 5. However, they could further elaborate on potential challenges in scaling their approach to high-dimensional or noisy data, and on the implications of using closed-source VLMs for reproducibility.

**Quality:**

3

**Strengths And Weaknesses:**

Strengths：
1.The paper presents a novel multi-modal, multi-step pipeline for automated model discovery, leveraging vision-language models to imitate human expert reasoning and visual judgment.
2.The introduction of the Visual Information Criterion to combine perceptual assessment with statistical metrics is an original and impactful contribution.
3.Experiments on time-series forecasting and symbolic regression demonstrate superior generalization performance over state-of-the-art baselines such as BoxLM and Automatic Statistician.

Weaknesses:
1.The current pipeline is demonstrated only on univariate (1D) datasets. While the approach shows promise, the visualization-based reasoning employed by EvaluatorVLM may not directly generalize to high-dimensional or multivariate data, where visual representation becomes ambiguous or infeasible. In such cases, the system would inevitably need to rely on numerical metrics or alternative modalities, raising questions about scalability.
2.Using VLMs to judge fitness and structure similarity is a novel and intriguing idea, but it also raises concerns about reliability. This is especially critical for extrapolated regions where human-like intuition from VLMs may not align with numerical accuracy. Appendix A.5 mentions averaging VLM assessments across two runs, but it is unclear how consistent these judgments are. How often do VLMs provide conflicting evaluations, and how does the system handle such cases?
3.The design of the Visual Information Criterion assumes that combining perceptual assessments from VLMs with BIC yields a balanced evaluation of model fitness and complexity. However, it is unclear how sensitive the final model selection is to the choice of the weighting parameter \alpha in VIC, or whether the method is robust to different trade-offs between these components. A deeper analysis or justification of this design choice would strengthen confidence in the approach.

---

> ### Author Rebuttal · Authors · 2025-07-31
>
> We thank reviewer fF4a for valuable comments and recognizing our work as:
>
> - **A novel multi-modal, multi-step pipeline for automated model discovery**, leveraging vision-language models to imitate human expert reasoning and visual judgment.
>
> - The introduction of the VIC to combine perceptual assessment with statistical metrics is **an original and impactful contribution.**
> - VLMs to judge fitness and structure similarity as **a novel and intriguing idea.**
>
> We have addressed concerns and comments in below.
>
> ### **Generalibility to high-dimensional & multivariate data**
>
> > *The current pipeline is demonstrated only on univariate (1D) datasets.*
> > *Do the authors have thoughts or preliminary experiments on how their approach could handle multivariate?*
>
> While a similar vein of research [13, 29, 32], including ours, has been only focused on the univariate data setting due to its importance and its broad applications, we agree that extending to the multi-variate case is an interesting direction.
>
> Given the limited rebuttal period, while we could not show a complete extension at this moment, we emphasize that our pipeline is not limited to the univariate case. Our proposed multi-modal multi-step pipeline is modular; thus, extending each module to multi-variate ones enables multi-variate data analysis. In this sense, we discuss the following potential directions.
>
> 1\) Since AnalyzerVLM conducts its analysis through code execution, AnalyzerVLM operation is **agnostic to the number of dimensions,** so applying it directly to multivariate data is possible. We conducted a preliminary experiment of AnalyzerVLM on multivariate data, and we found that AnalyzerVLM **checks the data shape** and **conducts the visualization per dimension**, and analyzes **pairwise interactions across dimensions**. Such analysis can be adopted per-dimension modeling or the adoption of similar models with highly correlated dimension, which is a good starting point of model discovery.
>
> 2\) For EvaluatorVLM, a simple and straightforward extension would be evaluating **each output dimension independently** and **aggregating scores**, similar to how mean squared errors are averaged across dimensions.
> As high-dimensional data modeling naturally incorporates the feature extraction to use good feature space \[C1, C2\] and enables finding interpretability at the feature space, we can adopt the new format of the plots (e.g., PCA/t-SNE, or feature space visualization) for EvaluatorVLM. So the format of plot may change for high-dimensional or multivariate datasets, but **the core idea of vision-based analysis and evaluation can be scaled up** upon our pipeline.
>
> \[C1\] Wilson et al., Deep Kernel Learning, Artificial intelligence and statistics. PMLR, 2016\.
> \[C2\] Alberto et al., Feature extraction and image processing for computer vision. Academic press, 2019\.
>
> ### **VLM assessments consistency**
>
> We appreciate the concern, especially on how VIC can be critical at extrapolate region, and visual assessment’s consistency.
>
> First, VIC is designed to regularize the traditional numerical metric with human intuition about data structure, allowing us to benefit from both traditional metrics and human intuition. As shown in Fig.9 in our paper, EvaluatorVLM’s evaluation shows a correlation with human evaluation, as well as the accuracy gap between interpolation and extrapolation regions, suggesting that the visual assessments capture not only numerical performance but also structural cues aligned with human intuition.
>
> Second, to assess the consistency of EvaluatorVLM, we conduct the experiments and report the standard deviation of the evaluations with 10 runs for models that are suggested during our pipeline. Since evaluatorVLM produces discrete scores on a 5-point scale in the range \[0, 50\], we find the variation to be within acceptable ranges (average of 3.11 and 5.44).
>
> |  | Airline | Solar | Mauna | Wheat | Temperature | Avg. |
> | :---- | :---- | :---- | :---- | :---- | :---- | :---- |
> | visual fitness | 2.4442 | 4.328 | 3.973 | 2.923 | 1.937 | 3.11 |
> | visual generalizability | 3.4509 | 1.951 | 7.042 | 6.200 | 3.815 | 5.44 |
>
> And we have checked top cases that cause the EvaluatorVLM to produce inconsistent judgments:
>
> 1) When the mean prediction doesn’t appear, the confidence interval covers all areas, EvaluatorVLM tends to be confused
> 2) When the data or prediction is too stiff, the evaluation varies
> 3) When the confidence interval expands at extrapolation (high uncertainty), but the mean prediction remains visually reasonable, EvaluatorVLM tends to assign low scores due to its uncertainty, and high scores for the mean predictions.
>
> To ensure stable and reliable assessment at such cases, we incorporated numerical evaluation into VIC, providing a consistent anchor point for visual scoring. In addition, we have averaged the scores across multiple runs (2 for our main experiments, but this averaging process can be scaled further if required), which stabilizes the assessments without incurring high computational costs.
>
> ### **Hyperparameter α selection & sensitivity**
>
> > *how sensitive the final model selection is to the choice of the weighting parameter $\alpha$ in VIC?*
>
> We set $\alpha$ so that $\alpha$ * EvaluatorVLM accounts for approximately 10-30% of the original metric (e.g., BIC), and performed grid search around this range to find the best $\alpha$.
> Since the scales of the selection criteria vary due to data normalization or criteria (e.g., BIC vs. NMSE) we set α such that the visual score contributes a consistent proportion to the total objective. As a result, α may seem vary significantly between tasks, but to maintain this relative weighting between visual and numerical components.
>
> We also provide the result of test RMSE of grid search varying α around 0 to 100, at Airline and Radio dataset. As shown, there is a certain turning point (or a break-down point) of alpha around 50. Also, we describe qualitative results: while α is 0, final searched models are dropping at extrapolated region, and when $\alpha$ increases, the pipeline tends to select the model with more generality even in high value ranges near 70. This hints that our hyperparameter setting is not sensitive and starting from introducing small α would be a good rule of thumb. We will add qualitative samples in Appendix.
>
>
> | alpha | 0 | 30 | 50 | 70 | 100 |
> | :---- | :---- | :---- | :---- | :---- | :---- |
> | Airline | 0.0937 | 0.0574 | **0.0534** | 0.0612 | 0.0824 |
> | Radio | 0.0766 | 0.0715 | **0.0562** | 0.0764 | 0.0954 |
>
> ### **Interpretability of formulas discovered by our pipeline**
>
> > *Could the authors comment on the interpretability of the symbolic regression formulas discovered by their method?*
>
> To demonstrate this, we report examples of formulas and discovered model:
>
> For Keijzer 3, the ground truth formula is $0.3x+\sin(2πx)$, and the discovered formula is $0.2963x+\sin(6.2781x)−0.001$, which closely matches the true function, both in structure and parameter values.
> We applied our method to the classical physics formula of free fall motion: $y(t)=v_0t-\frac{1}{2}gt^2$, while $v_0=10$ and $g=9.81$. And the discovered final formula is $-4.9065x^2 + 10.0164x - 0.0042$, effectively showing that our pipeline can successfully recover the interpretable equation from data.
>
> We also provide the discovered kernel structure interpretation of the Radio dataset (Fig. 2 in our paper), which our discovered model $\text{PER} \* (\text{PER} \+ \text{SE})$ revealed two periodic components: 10.25 years for the first period, 1.01 years for the second period, which alignes the data’s characteristics with multiple periods that the dataset has.
>
> The formulas and models discovered by our pipeline are not only compact but also structurally interpretable, often aligning with forms that are familiar and meaningful to human experts, highlighting that our method is effective in producing expressions that are both accurate and explainable.

---

> > ### Comment · Reviewer_fF4a · 2025-08-05
> >
> > the authors more or less address my comments, but I still have concern on the reliability to use VLMs to judge fitness and structure similarity. In your new experiment, do you mean the score range is [0 5] since it is a 5-point evaluation, and why did you only perform 10 runs to compute the variance?

---

> ### Author Response · Authors · 2025-08-06
> **Clarification on New Experiment**
>
> To clarify, we provide the experiment details and additional analysis.
>
> First, the range of score is in [0, 50], i.e., {0, 5, 10, 15, 20, 25, 30, 35, 40, 45, 50}.
>
> And the statistics were mis-conveyed. We apologize.
> To be clear,  for each model, we computed the standard deviation (std) of both visual fitness and visual generalizability scores by repeating 10 times.
> To measure the consistency, we report the **mean of std and its 95% confidence interval (CI) across models** as follows:
>
> **Visual Fitness Consistency** (average std ± 95% CI):
> - Overall: **3.11 ± 1.87**
> - Airline (37 models): 2.44 ± 1.52
> - Solar (23 models): 4.33 ± 2.36
> - Mauna (34 models): 3.97 ± 2.46
> - Wheat (42 models): 2.92 ± 1.82
> - Temperature (36 models): 1.94 ± 1.20
>
> **Visual Generalizability Consistency** (average std ± 95% CI):
> - Overall: **5.44 ± 2.92**
> - Airline (37 models): 3.45 ± 2.14
> - Solar (23 models): 1.95 ± 1.21
> - Mauna (34 models): 7.04 ± 4.37
> - Wheat (42 models): 6.20 ± 3.84
> - Temperature (36 models): 3.82 ± 2.36
>
> Specifically, for the "Overall" experiment, we collected **172 models across 5 datasets and evaluated each model 10 times, resulting in 1720 runs in total.**
> These models were not arbitrarily sampled, they are the whole set of candidate models generated during the execution of our pipeline.
>
> These results suggest that **10 evaluation runs per model provide a reasonably stable estimate of EvaluatorVLM’s consistency**. While some variability is naturally expected due to the stochastic nature of VLM outputs, the measured standard deviations remain within reasonable intervals across datasets, i.e., consistent.

---

> > ### Author Response · Authors · 2025-08-08
> > **Further Discussion**
> >
> > We appreciate reviewer fF4a, for taking the time to review our submission and providing such constructive feedback.
> >
> > If there are any parts of our rebuttal that remain unaddressed, please feel free to request a discussion, we’d be happy to respond. Additionally, we’d be happy to answer any further questions you may have.
> >
> > Best regards,
> >
> > Authors

---

### Official Review · Reviewer_Nkkg · 2025-07-03

**Clarity:** 3
**Significance:** 3
**Originality:** 3
**Rating:** 5
**Confidence:** 4

**Summary:**

This work levrages two Vision-Language Models (AnalyzerVLM + EvaluatorVLM) to automatically discover mathematical models for time series data. AnalyzerVLM iteratively analyzes data through code generation, EvaluatorVLM evaluates models using a novel Visual Information Criterion (VIC) that combines traditional metrics with visual assessment.

**Questions:**

How does this work with high-dimensional data where simple plots aren't possible?
How do you choose α (varies 1000x between applications) without extensive tuning?
When does visual assessment give wrong answers? What about noisy/counter-intuitive data?
How expensive is multi-step VLM inference really? Is this practical at scale?

**Ethical Concerns:**

["NO or VERY MINOR ethics concerns only"]

**Final Justification:**

Thank you for rebuttal. Increased my score.

**Limitations:**

See Weaknesses above.

**Quality:**

3

**Strengths And Weaknesses:**

Strengths

+ Intresting to use VLMs for automated model discovery with human-like visual reasoning
+ Outperforms baselines (ARIMA, Prophet, etc.) across different datasets with better generalization
+ Captures human intuition about model reasonableness through visual evaluation

Weaknesses

- Only 1D time series was tested
- Multi-step VLM inference could be expensive, unclear how it scales
- VIC weighting (α) and scoring thresholds lack principled selection

---

> ### Author Rebuttal · Authors · 2025-07-31
>
> We thank reviewer Nkkg for valuable comments, and recognizing our work as:
>
> - Interesting to use **VLMs for automated model discovery** with human-like visual reasoning
>
> - **Novel Visual Information Criterion (VIC)** that combines traditional metrics with visual assessment
>
> - **Capturing human intuition** about model reasonableness through visual evaluation
>
> We have addressed your concerns and comments below.
>
> ### **Extension to high-dimensional data & noisy data**
>
> > *Only 1D time series was tested.*
>
> > *How does this work with high-dimensional data where simple plots aren't possible?*
>
> While a similar vein of research [13, 29, 32], including ours, has been only focused on the univariate data setting due to its importance and its broad applications, we agree that extending to the multi-variate case is an interesting direction.
>
> Given the limited rebuttal period, while we could not show a complete extension at this moment, we emphasize that our pipeline is not limited to the univariate case. Our proposed multi-modal multi-step pipeline is modular; thus, extending each module to multi-variate ones enables multi-variate data analysis. In this sense, we discuss the following potential directions.
>
> First, AnalyzerVLM can be directly applied to multivariate data without any architectural modifications. As it conducts its analysis through data access code snippets, no additional resources (e.g., tokens) are required to handle multivariate inputs. Based on this, we have conducted additional experiments of AnalyzerVLM at multivariate data, and we observed that AnalyzerVLM conducts coordinate visualization, and pairwise interactions analyses, drawing correlations across dimensions. Such analyses from AnalyzerVLM can inform per-dimension modeling of the data or the adoption of similar models with highly correlated dimensions, providing a strong foundation for the model discovery process.
>
> Second, EvaluatorVLM visually evaluates the resemblance between the prediction and the data. A simple and straightforward extension would be to evaluate algorithmically (e.g., evaluate each output dimension independently and aggregate scores, or evaluate randomly sampled dimensions alternatively according to each iteration, like randomized algorithms), similar to how mean squared errors are averaged across dimensions in conventional scenarios.
>
> In addition, for high-dimensional data, one can adopt alternative plot formats incorporating joint decompositions (e.g., PCA, t-SNE). While the visualization format may change for high-dimensional or multivariate datasets, the core idea of vision-based analysis and evaluation can be scaled up upon our pipeline.
>
> We appreciate the reviewer’s suggestion regarding high-dimensional extensions; it points to a valuable direction, and we believe further exploration along these lines would be a meaningful next step to broaden our approach.
>
>
> > *What about noisy/counter-intuitive data?*
>
> To evaluate our pipeline’s robustness, we conducted experiments on the Airline, Mauna datasets (which has the increasing trend with periodicity (shapes in Fig. 11)) by introducing controlled Gaussian noise (σ = 0.03~0.05).
> As long as the noise did not completely override the underlying trends in the data (σ > 0.5), the final searched model was able to capture the trend of data, showing our method’s robustness. Below we report test RMSE results with the discovered structure (LIN: linearity, PER: periodicity, SE: locality) The results indicate that the pipeline maintains reasonable performance under moderate noise levels, consistently capturing core patterns in the data. We also observed that as the data becomes noisier, the identified models more frequently incorporate local components (e.g., SE), suggesting a tendency to reflect finer-scale patterns in noisier settings. This may hint that local components like SE absorb the noise or outlier factors, and it could be exploited to deal with noisy or outlier cases by post-processing those.
>
> | noise level | σ=0 | σ=0.03 | σ=0.05 |
> | :---- | :---- | :---- | :---- |
> | Airline (RMSE) | 0.0534 | 0.0864 | 0.0725 |
> | Airline (Model) | LIN \* (PER \+ SE) | LIN \* PER \+ SE | SE \* (PER \+ C) |
> | Mauna (RMSE) | 0.0564 | 0.0656 | 0.1073 |
> | Mauna (Model) | (SE \+ LIN) \* (PER \+ C) | SE \* PER \+ C | (SE \+ C) \* (PER \+ C) |
>
> ### **Cost of Multi-step VLM inference**
>
> > *Multi-step VLM inference could be expensive, unclear how it scales*
>
> > *How expensive is multi-step VLM inference really? Is this practical at scale?*
>
> Surprisingly, it is not expensive compared to [C1, C2].
> To ensure how multi-step VLM inference consumes, we have calculated the cost of multi-step inference in our pipeline. The result shows that 1) visual representation of data reduces each step’s token usage 2) multi-step inference has the effect of reducing the whole round, leading to a whole cost reduction. So such characteristics of multi-step VLM inference reduces the total cost of our pipeline, making it more efficient.
>
> First, our data encoding with plots of AnalyzerVLM is token-independent, making the token usage the same regardless of the number of data points. In contrast, the LLM-based discovery frameworks [C1, C2] often utilize data directly into textual prompts, which leads to the excessive use of tokens.
> For example, to encode around 500-1000 points, AnalyzerVLM incorporates images to 1045 tokens, while LLM based encoding takes around 8000 tokens per round. This not only rises token usage but also limits scalability when dealing with longer sequences.
>
> Second, for the scalability of the multi-step pipeline, our AnalyzerVLM takes the step of 6 per round on average, with a total 5 rounds through the input of 1,500 tokens including both text and image tokens per step. Therefore, the total token consumption can be approximated at around 45,000 tokens (\\$0.03 with GPT-4o-mini, \\$0.25 with GPT-4o in total). Compared to the LLM-based model discovery [C1, C2], which relies on heavy iterations(10~50 rounds) with one-step generation may require around 8000*10=80000 tokens, our methods effectively generate a reliable model, making an effect of reducing overall cost.
>
> Our method incorporates VLM with multi-step reasoning, which can lead to the reduction of total rounds through multi-step reasoning, and the vision-based encoding of the data to make the number of tokens.
>
> |  | token usage per step | Number of steps | Number of Rounds | Total token usage |
> | :---- | :---- | :---- | :---- | :---- |
> | One-step LLM (LLM as optimizer [C4]) | 8000 | 1 | \~10 | 80,000 (\$0.053) |
> | Ours (Multi-step VLM) | 1500 | 6 | 5 | 45,000 (\$0.030) |
>
> \[C1\] Yang et al., Large Language Models as Optimizers, ICLR 2024\.
> \[C2\] Merler et al., In-Context Symbolic Regression: Leveraging Large Language Models for Function Discovery, ACL 2024\.
>
> ### **Hyperparameter selection principle for VIC weighting**
>
> > *VIC weighting (α) and scoring thresholds lack principled selection*
>
> > *How do you choose α (varies 1000x between applications) without extensive tuning?*
>
> We set $\alpha$ so that $\alpha$ * EvaluatorVLM accounts for approximately 10-30% of the original metric (e.g., BIC), and performed grid search around this range to find the best $\alpha$.
> Since the scales of the selection criteria vary due to data normalization, we set α such that the visual score contributes a consistent proportion to the total objective. As a result, α may seem vary significantly between tasks, but to maintain this relative weighting between visual and numerical components.
>
> We also provide the result of test RMSE of grid search varying α around 0 to 100, at Airline and Radio dataset. As shown, there is a certain turning point (or a break-down point) of alpha around 50. Also, we describe qualitative results: while α is 0, final searched models are dropping at extrapolated region, and when $\alpha$ increases, the pipeline tends to select the model with more generality even in high value ranges near 70. This hints that our hyperparameter setting is not sensitive and starting from introducing small α would be a good rule of thumb. We will add qualitative samples in Appendix.
>
> | alpha | 0 | 30 | 50 | 70 | 100 |
> | :---- | :---- | :---- | :---- | :---- | :---- |
> | Airline | 0.0937 | 0.0574 | **0.0534** | 0.0612 | 0.0824 |
> | Radio | 0.0766 | 0.0715 | **0.0562** | 0.0764 | 0.0954 |
>
> ### **When does visual assessment give wrong answers?**
>
> We have observed the cases that the visual assessment of our VIC gives wrong answers:
>
> -  When data/prediction line is too stiff
>
> - When the mean prediction is out of the plot range and the confidence interval fills the whole plot
>
> - When the confidence interval grows at each side (high uncertainty at OOD), but the mean prediction shows a maintained structure
>
> To ensure stable and reliable assessment in such cases, we incorporated numerical evaluation into VIC, providing a consistent anchor point for visual scoring. In addition, we have averaged the scores across multiple runs (2 for our main experiments, but this averaging process can be scaled further if required), which stabilizes the assessments without incurring high computational costs.
>
> Searching for a better plot format for stiff data, may enhance the performance. We believe future work for adaptive plot formatting strategies,: e.g., scaling y-axis, emphasizing residuals, or visualizing derivatives, can be adopted to our pipeline to enhance structure visibility.

---

> ### Author Response · Authors · 2025-08-06
> **Addressing Remaining Concerns**
>
> We sincerely appreciate reviewer Nkkg’s time and effort in reviewing our submissions and providing valuable feedback.
> If there are remaining concerns, we would be happy to provide any additional clarification or engage in further discussion. Please feel free to initiate further discussion during the discussion phase.

---

> ### Author Response · Authors · 2025-08-08
> **Further Discussion**
>
> We appreciate reviewer Nkkg, for taking the time to review our submission and providing such constructive feedback.
>
> If there are any parts of our rebuttal that remain unaddressed, please feel free to request a discussion, we’d be happy to respond. Additionally, we’d be happy to answer any further questions you may have.
>
> Best regards,
>
> Authors

---

### Author Response · Authors · 2025-08-09
**Dear Area Chair and Reviewers**

We thank Area Chair and all Reviewers for their time and dedication for review process, and for the constructive discussions. We deeply value their insights, which have helped us to better articulation of its contributions.

We emphasize that our contribution is to demonstrate that **statistical modeling of data can be automated by leveraging the capability of VLMs**. Focusing on the key contribution in this work, multivariate modeling is a completely separate line of research that tackles framework-level contribution, c.f., relational kernel learning [C1, C2], while we have contributed on uncovering the capability of our AnalyzerVLM and EvaluatorVLM followed by the integrated system and evaluating and demonstrating the VLM’s capabilities for statistical modeling.

Although we do not focus on tackling multi-variate cases in this work, we believe that our developments and findings can be extended to multivariate cases with such extensions, e.g., coordinate descent like procedures; thus, focusing on the 1D case should not be confused as our inherent limitation, but a different scope. We highlight this clear difference.

We **thank Area Chair and Reviewers again for their encouraging feedback and dedication**, and trust that this clarification highlights the intent and novelty of our contributions, as well as their potential for broader impact through future extensions.


[C1] Nguyen et al., Heterogeneous Relational Kernel Learning, MileTS '19

[C2] Hwang et al., The Automatic Statistician: A Relational Perspective, ICML 2016

---

### Note · Authors · 2025-08-13

## Dear Area Chair and All Reviewers
We appreciate the Area Chair for coordinating the review process of our work, and thank all Reviewers for their constructive and engaging discussions, and recognizing our work as:
- **Interesting and novel multi-modal, multi-step pipeline for automated model discovery** (Reviewer Nkkg, fF4a, MXP2)
- **Novel VIC** to combine statistical metrics with visual assessment, which **an original and impactful contribution** (Reviewer Nkkg, fF4a)
- **Capturing and imitating human intuition** for model discovery (Reviewer Nkkg, fF4a, MXP2)
- **Generalization performance on multiple datasets and baselines** (Reviewer fF4a, MXP2)

We have addressed all the reviewers' comments and had discussions for:
- **Multivariate & high dimensional data (Reviewer Nkkg, fF4a)**: We clarified that pipeline is modular and extendable; provided preliminary multivariate possibility and visualization strategies.
- **Hyperparameter $\alpha$ selection sensitivity (Reviewer Nkkg, fF4a)**: We explained weighting principle, sensitivity results of $\alpha$, and qualitative effects.
- **Ablation study of different plot formats (Reviewer MXP2)**: We reported how the visual assessment varies according to the plot's resolution, line thickness, and line color.
- **Visual assessment (Reviewer Nkkg, fF4a)**: We reported the standard deviation of 10 repeated visual assessments per model, and the top cases with the largest variability.
- **Qualitative interpretation examples of the discovered formula/models (Reviewer fF4a, MXP2)**: We reported the examples of the discovered models with their parameters, showing that they align with the data's characteristics.
- **Additional clarifications and enhancements**: We clarified the cost of multi-step VLM Inference (Reviewer Nkkg), AnalyzerVLM's generated code examples (Reviewer MXP2), methods for balancing interpretability and model fit (Reviewer MXP2),  and conducted clarification of Figures 2 and 3 (Reviewer MXP2).

We sincerely thank the Area Chair and all reviewers for their insightful feedback and constructive suggestions, which have helped us strengthen our work. We engaged in meaningful discussions with the reviewers and believe we have satisfactorily addressed the majority of their concerns.

---

### Decision · Program_Chairs · 2025-09-17

**Decision:**

Accept (poster)

**Comment:**

This paper takes a creative step by using VLMs as both analyzers and evaluators for automated model discovery. The Visual Information Criterion is a particularly interesting idea with human-like intuition. The reviewers agreed the work is technically solid, with promising results. Concerns around scaling to multivariate data, consistency of VLM-based judgments, and interpretability were discussed in the rebuttal with a few new small experiments. The work has clear limitations in scope but it also opens up an exciting direction worth accepting.